# A multi-regional human brain atlas of chromatin accessibility and gene expression facilitates promoter-isoform resolution genetic fine-mapping

Pengfei Dong [1,2,3,4] ✉, Liting Song[1,2,3,4], Jaroslav Bendl [1,2,3,4], Ruth Misir[2,3,4], Zhiping Shao[1,2,3,4], Jonathan Edelstien[2,3,4], David A. Davis[5], Vahram Haroutunian [2,3,6,7], William K. Scott [5,8], Susanne Acker[9], Nathan Lawless [9], Gabriel E. Hoffman [1,3,4,7,10], John F. Fullard [1,2,3,4] & Panos Roussos [1,2,3,4,7,10] ✉

Brain region- and cell-specific transcriptomic and epigenomic features are associated with heritability for neuropsychiatric traits, but a systematic view, considering cortical and subcortical regions, is lacking. Here, we provide an atlas of chromatin accessibility and gene expression profiles in neuronal and non-neuronal nuclei across 25 distinct human cortical and subcortical brain regions from 6 neurotypical controls. We identified extensive gene expression and chromatin accessibility differences across brain regions, including variation in alternative promoter-isoform usage and enhancer-promoter interactions. Genes with distinct promoter-isoform usage across brain regions were strongly enriched for neuropsychiatric disease risk variants. Moreover, we built enhancer-promoter interactions at promoter-isoform resolution across different brain regions and highlighted the contribution of brain region-specific and promoter-isoform-specific regulation to neuropsychiatric disorders. Including promoter-isoform resolution uncovers additional distal elements implicated in the heritability of diseases, thereby increasing the power to fine-map risk genes. Our results provide a valuable resource for studying molecular regulation across multiple regions of the human brain and underscore the importance of considering isoform information in gene regulation.

Interpreting the functional consequences of non-coding genetic risk variants associated with neuropsychiatric disorders such as schizophrenia (SCZ)[1], and bipolar disorder (BD)[2] presents a significant challenge[3,4]. These variants are enriched in cis-regulatory elements (CREs) of the central nervous system (CNS)[5–12], linking them to their target genes and understanding their impact on gene regulation remains elusive. Promoter-isoforms, which are alternative transcription start sites controlling the expression of different gene isoforms[13,14], serve as the fundamental units of transcriptional regulation[15] and enhancer-promoter (E-P) interactions. But a systematic understanding of enhancer-promoter interactions at promoter-isoform resolution is lacking, hindering our ability to fine-map genetic associations in neuropsychiatric disorders.

The human brain is a highly complex organ consisting of myriad cell types that reside in specific brain regions that are associated with different cognitive functions. Consequently, the gene expression and epigenome landscapes can vary dramatically between brain regions and cell types[8,16,17]. While previous efforts focused on the cortical areas of the brain, the molecular mechanisms regulating subcortical areas, including midbrain (MidBr), diencephalon (Dien), and basal ganglia (BasGan) are largely unknown. Emerging evidence from genetic association[8,16,18] and neuroimaging[19,20] studies suggest a critical role for subcortical areas in neuropsychiatric disorders. One prominent example is BasGan-specific medium spiny neurons (MSN), in which both expressed genes[1,18] and accessible chromatin[8,16] are strongly enriched for common SCZ risk variants, and are independent of other neuronal cell populations[18]. We previously profiled chromatin accessibility across cortical and subcortical areas and highlighted the function of striatum CREs in neuropsychiatric disorders[8]. However, the previous data set included only a limited number of subcortical areas and lacked coupled gene expression information. Thus, the cell-type-specific promoter- and enhancer- regulatory landscape across brain regions, as well as the association with risk variants in neuropsychiatric disorders, remain uncharacterized.

To address these gaps, we comprehensively profiled gene expression and chromatin accessibility in neuronal and non-neuronal nuclei across 25 brain regions from 6 individuals with no history of neuropsychiatric or neurodegenerative disease. Our analysis revealed extensive regional differences, particularly within subcortical areas, underscoring the importance of incorporating multiple brain regions in studies of neuropsychiatric disorders. We discovered alternative promoter-isoform usage across different brain regions and examined E-P links at promoter-isoform resolution. We found that enhancers linked specifically to non-5′ promoter-isoforms contribute significantly to neuropsychiatric disorders and enhance the fine-mapping of risk genes. Our data provide a valuable resource for understanding the regulatory mechanisms underlying neuropsychiatric traits and emphasize the critical role of promoter-isoforms and regional specificity in genetic fine-mapping.

## Results

### An atlas of chromatin accessibility and gene expression in neuronal and non-neuronal nuclei across 25 brain regions

To comprehensively characterize gene expression and chromatin accessibility across cortical and subcortical structures, we generated matched ATAC-seq and RNA-seq profiles in neuronal and non-neuronal nuclei isolated from 25 brain regions covering the forebrain (ForeBr), BasGan, midbrain/diencephalon (MidDien) and hindbrain (HinBr) of six control subjects (Fig. 1a, b, Supplementary Fig. 1, Supplementary Table 1, and Supplementary Data 1–3). After rigorous quality control, including assessing cell type, sex, genotype concordance, and quality metrics (Supplementary Figs. 2–7), we obtained a total of 14.2 billion uniquely mapped paired-end read pairs for RNA-seq ($N = 265$), and 15.1 billion uniquely mapped paired-end read pairs for ATAC-seq ($N = 202$). We examined the abundance of neuronal brain region marker genes, including *KCNS1* (ForeBr), *DRD2* (BasGan), *IRX3* (MidDien), and *GABRA6* (HinBr), and found concordant brain region specificity for both gene expression (Supplementary Fig. 8a) and chromatin accessibility (Fig. 1c). We generated an atlas of open chromatin regions (OCRs) by calling peaks across 25 brain regions for each cell type (Supplementary Fig. 8b), yielding 320,308 and 196,467 neuronal and non-neuronal peaks, respectively. To further confirm the brain-region and cell-type specificity of OCRs, we compared our results with a recent multi-brain-region single-cell ATAC-seq dataset[16]. In line with the previous analyses[8,16], both neuronal and non-neuronal OCRs were enriched in their corresponding cell types, and neuronal OCRs exhibited brain region specificity (Supplementary Fig. 8c), as previously reported.

### Consistent transcriptomic and chromatin accessibility differences across brain regions

To visualize the genome-wide molecular feature similarities between cell type and brain regions, we applied Principal component analysis (PCA) to both gene expression and chromatin accessibility (Supplementary Fig. 8d). Consistent with our previous analysis, samples clustered first by cell type and next by brain region in both assays[8]. We performed differential analysis for genes and OCRs for each pair of brain regions and used Storey's $\pi_1$ statistic[21] as a metric of dissimilarity across brain regions (Fig. 1d). In neurons, both assays exhibit robust differences across broad brain regions that include ForeBr, BasGan, MidDien, and HinBr. In non-neurons, MidDien exhibits robust differences compared to the other broad brain regions (Fig. 1d and Supplementary Fig. 8e). The HinBr exhibits the most significant differences in comparison to the other brain regions. However, given the challenge of separating neuronal from non-neuronal nuclei in HinBr[22], we provided HinBr RNA-seq and ATAC-seq as a resource but did not include these data in downstream analyses. Moreover, within the broad brain regions, there is a substantial distinction observed within MidBr (rostromedial tegmental nucleus, RMTG; ventral tegmental area, VTA; dorsal raphe nucleus, DRN) and Dien (arcuate nucleus, ARC; habenula, HAB; mediodorsal thalamus, MDT) in both neuronal and non-neuronal cells, while the limbic areas (amygdaloid complex, AMY; hippocampus, HIPP) displayed dissimilarities compared to the neocortex (NEC) specifically in neuronal cells (Supplementary Fig. 8e), indicating the cellular diversity within such areas.

To determine the relationship between the two molecular features, we assessed the genome-wide consistency between gene expression and the corresponding chromatin accessibility. Consistent with previous transcriptional regulation models[23–25], we observe a highly significant correlation between gene expression and promoter chromatin accessibility (Supplementary Fig. 8f). To further assess the shared correlation structure between RNA-seq and ATAC-seq, we performed canonical correlation analysis (CCA)[26] and found the samples separated by cell type and brain region, instead of assay (Fig. 1e), demonstrating the consistency of both approaches.

We found that, in neurons, 87.6% (19,157/21,878) of genes and 79.5% (235,722/296,337) of OCRs are significantly differentially expressed/accessible in at least one of the pairwise comparisons between ForeBr, BasGan, and MidDien (global FDR < 0.05) (Supplementary Data 4–6). We took a conservative approach and considered genes/OCRs as broad-brain-region specific only if they were significantly more expressed/accessible in all pairwise comparisons against the remaining broad-brain-regions. Consistent with our previous analysis[8], BasGan has the highest number of differentially expressed genes (DEGs) and differentially accessible chromatin regions (DACs) (Supplementary Fig. 9a). Interestingly, neuronal BasGan-specific genes contain a significantly higher fraction of non-coding RNAs (Supplementary Fig. 9b). One such example is the *MALAT1* gene, which has been implicated in multiple neuropsychiatric traits[27] (Supplementary Fig. 9c).

To further examine the brain-region and cell-type specificity of our findings, we compared our differential analysis with two independent multi-brain-region single-cell reference sets[16,18]. Consistent with known brain region anatomy, the neuronal DEGs from ForeBr, BasGan, and MidDien were enriched for pyramidal cells, MSN, and MidBr/hypothalamus neurons, respectively (Fig. 1f). The DACs exhibit similar cell-type associations (Fig. 1f). Within the ForeBr in neurons, DEGs and DACs of NEC and limbic areas were strongly enriched for their corresponding excitatory neurons (Fig. 1f and Supplementary Fig. 10a), whereas limbic DEGs were also enriched for HIPP specific inhibitory neuron signatures. This latter observation is consistent with recent findings indicating that inhibitory neurons are similar between NEC and limbic areas, while excitatory neurons are highly distinct[28]. As expected, MidBr-specific DEGs were enriched for MidBr neurons and

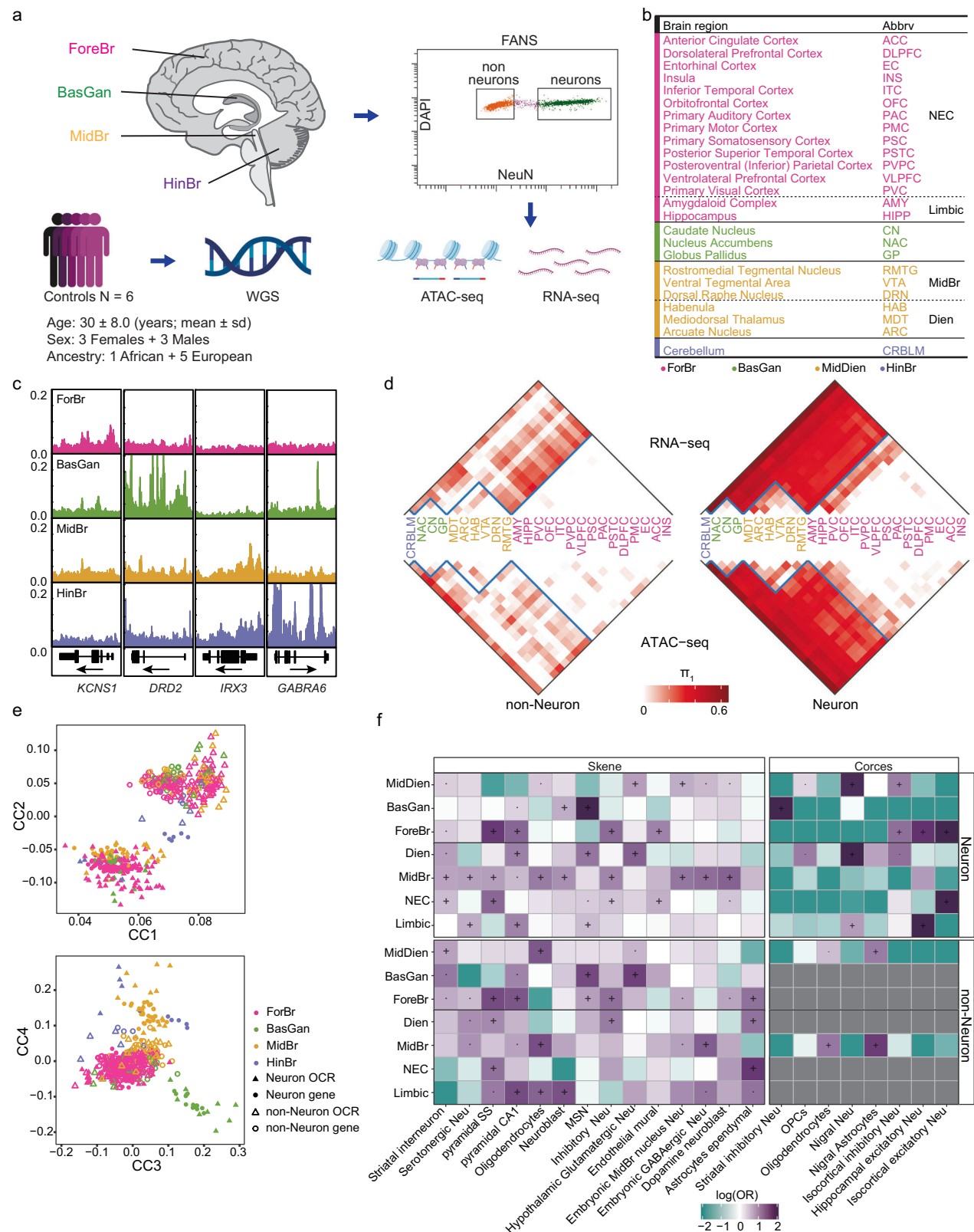

dopamine neurons, while Diens were enriched for hypothalamus neurons. However, we did not observe significant enrichment of MidBr DACs, which is consistent with the absence of concordant brain regions in the reference. In non-neurons, MidDien, MidBr, and limbic DEGs were enriched for oligodendrocytes (ODC), while ForeBr, Dien, and NEC DEGs were enriched for astrocytes, suggesting variations in

cell composition across different brain regions (Fig. 1f). Furthermore, the MidDien DACs were also enriched for Nigra astrocytes (Fig. 1f), consistent with a recent report of astrocyte heterogeneity between the MidDien and other brain regions[29]. It's worth noting that, in contrast to patterns of chromatin accessibility, DEGs in non-neuronal cells showed enrichment for markers (Fig. 1f) and pathways (Supplementary

**Fig. 1 | Extensive and consistent gene expression and chromatin accessibility across 25 human brain regions. a** Schematic representation of the study design. Postmortem samples were dissected from 25 brain regions across ForeBr, MidDien, BasGan, and HinBr. Nuclei were subjected to fluorescence-activated nuclear sorting (FANS) to yield neuronal (NeuN+) and non-neuronal (NeuN-) nuclei, followed by ATAC-seq and RNA-seq profiling. We also performed whole-genome sequencing for each individual. Sd represents standard deviation. Schematic was created using BioRender (https://biorender.com). Fullard, J. (2021) BioRender.com/h19m968, Fullard, J. (2021) BioRender.com/p82x857, and Fullard, J. (2021) BioRender.com/w09a902. **b** The 25 brain regions and abbreviations (anatomy dissection in Supplementary Fig. 1). **c** Chromatin accessibility profiles merged from the neuronal broad brain regions around brain region-specific marker genes. **d** Pairwise statistical dissimilarity (quantified based on the proportion of true non-null tests, π1) across different brain regions of neuronal and non-neuronal cells in the two assays. **e** the shared correlation structures (canonical correlation vectors, CC) between ATAC-seq and RNA-seq are separated by cell type (top, CC1-2) and brain regions (bottom, CC3-4). **f** Cell type enrichment determined by one-tailed Fisher's exact test in brain region-specific gene (for DEG)[18] and OCR (for DAC)[16] sets. Groups with a low number of significant differential results were masked and colored gray. Neu neuron. pyramidal SS somatosensory pyramidal cells. OPC oligodendrocyte progenitor cells. Odds ratio (OR). "-": Nominally significant (P < 0.05); "+": significant after FDR (Benjamini & Hochberg) correction (FDR < 0.05). Source data is provided as a Source Data file.

Fig. 10b) characteristic of neuronal cells from the corresponding brain regions. This observation could be attributed to the presence of ambient RNAs[30] coupled with the high degree of regional homogeneity in non-neuronal cells. Consequently, subsequent analyses specifically focused on neuronal cells.

We further assessed the association between DEGs/DACs and common neuropsychiatric disorder-associated variants (Methods). Consistent with the previous analysis, neuronal brain region-specific molecular features were robustly associated with multiple neuropsychiatric traits (Supplementary Fig. 10c).

### Transcriptome and chromatin accessibility are highly heterogeneous across subcortical areas

Having shown broad brain region differences, we next delved into the molecular heterogeneity across more specific regions. Despite its larger volume, the NEC displayed limited molecular changes compared to other brain regions (Fig. 1d, Supplementary Figs. 8e, and 11a). Primary visual cortex (PVC) exhibits the most significant molecular differences and is enriched for excitatory neurons (Fig. 2 and Supplementary Fig. 11b), in line with previous studies highlighting the distinct nature of excitatory neurons in PVC compared to other regions[31]. Additionally, we identified gene expression changes in the entorhinal cortex (EC), with DEGs showing similarities to those found in limbic areas (Fig. 2a and Supplementary Fig. 11c), in accordance with their anatomical position. As expected, HIPP-specific DEGs were enriched for specific excitatory neurons characteristic of the hippocampus (Fig. 2b and Supplementary Fig. 11b).

In subcortical areas, we observed a higher degree of molecular changes across the fine brain regions. Notably, striatum (NAC) and the pallidum (GP) within BasGan exhibited substantial differences, with negatively correlated DEGs indicating molecular distinctions between the NAC and GP within BasGan (Supplementary Fig. 11c). Furthermore, the GP DEGs were enriched for synaptic function and GABA B receptor activation pathway, in keeping with the inhibitory effects of this region[32] (Fig. 2b). Within MidBr, consistent with its function, the VTA DEGs showed enrichment for dopamine neuron markers. Notably, the DACs specific to the VTA showed an enrichment of genetic risk variants associated with major depression (Supplementary Fig. 11d). This aligns with prior analyses highlighting the role of dopamine neurons in the pathology of depression[33]. In diencephalon, our analysis included structures such as the hypothalamus (ARC), subthalamus (MDT), and epithalamus (HAB). As expected, the ARC DEGs were enriched for previously defined hypothalamus neuron cell makers (Fig. 2b). The HAB region showed enrichment for cholesterol biosynthesis pathway, which aligns with the role of pineal gland neurons in producing neurosteroids[34] (Fig. 2b). MDT was enriched for excitatory neuron markers (Fig. 2), and was strongly associated with neuropsychiatric genetic risk variants (Supplementary Fig. 11d). Furthermore, we observed consistency between the DEGs and DACs (Supplementary Fig. 11e). Many of the brain regions examined in this study have not been extensively studied previously, and our findings indicate that fine brain areas within subcortical regions exhibit more pronounced differences in gene expression and chromatin states than those within cortical regions.

### Alternative promoter-isoform usage across brain regions identified neuropsychiatric disorder susceptible gene sets

Usage of alternative promoters regulates isoform usage pre-transcriptionally[13,14]. Promoter-isoform expression can be quantified by examining the set of unique junction reads transcribed from the promoter using RNA-seq data[35]. As full-length isoforms significantly outnumber promoter-isoforms, the quantification of promoter-isoform expression is substantially more robust[35]. By modeling the junction reads that are uniquely identifiable for the first exon (Fig. 3a, Supplementary Data 7, and Methods), we detected 13,108 uniquely identifiable promoter-isoforms (9108 5′ promoters and 4000 non-5′ promoters) from 11,224 genes. Although without regulatory genomics annotation, the 5′ promoter is often assumed as the default promoter, we found, at least for a fraction of genes (1344/11,224), that the 5′ promoter-isoform is not the most highly active (i.e., major promoter, Supplementary Fig. 12a). To validate our annotation of the non-5′ major promoters, we utilized independent promoter-associated epigenomic data (including ATAC-Seq, and H3K4me3 and H3K27ac ChIP-Seq), and, as expected, the major promoters (non-5′) exhibit more potent active epigenomic modifications than the 5′ promoters (Supplementary Fig. 12b).

To examine promoter-isoform-specific brain region alterations, we compared neuronal DEGs between promoter-isoform and the parent gene across brain regions. As expected, the majority of promoter-isoforms are concordant with the parent genes (Supplementary Fig. 12c, and Supplementary Data 8, 9). A fraction of promoter-isoforms is significantly differentially expressed between brain regions (FDR < 0.05), while the parent gene is either not significant (nominal P > 0.1), or shows the opposite direction (promoter-isoform specific DEG, Supplementary Fig. 12c and Methods), suggesting an alternative promoter-isoform usage in both broad brain regions and fine brain regions. In contrast to the majority of genes (Supplementary Figs. 8f and 11e), the parent gene expression of these promoter-isoforms exhibits a very limited association with OCRs at the 5′ promoter (Fig. 3b; top panel); while promoter-isoform expression exhibits a substantially higher correlation with promoter OCRs across brain regions (Fig. 3b; bottom panel), providing additional support for using promoter-isoform, rather than gene expression, to explore transcriptome differences across brain regions. One prominent example is the doublecortin like kinase 1 (*DCLK1*) gene, an essential regulator of synaptic development and axon regeneration that consists of two promoter-isoforms (a long and short form, respectively, Fig. 3c)[36]. The two promoter-isoforms have distinct functions in synapse development[37]. The long isoform is highly expressed in ForeBr and has the lowest expression in BasGan. In contrast, the short isoform is highly expressed in BasGan. Consistent with this, the promoter OCR of the short isoform also has higher chromatin accessibility in BasGan (Fig. 3c). Moreover, the promoter-isoform also exhibits alternative usage between limbic and NEC, and between MDT and Dien.

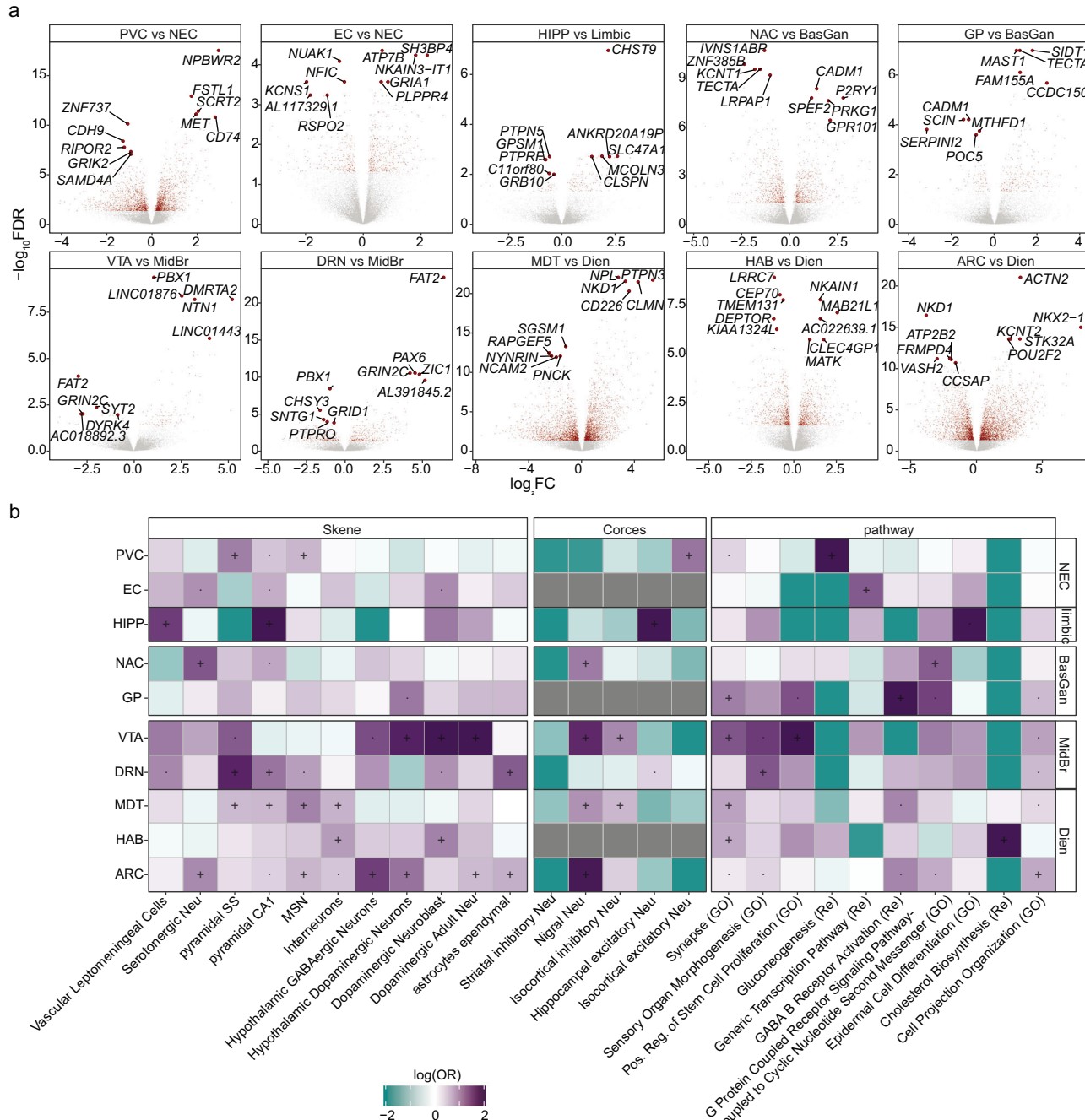

**Fig. 2 | Neuronal gene expression and chromatin accessibility changes in fine brain regions. a** Volcano plot of RNA-seq data from fine brain regions in neuronal cells. Significant genes after FDR (Benjamini & Hochberg) correction (FDR < 0.05) are marked in red. The top 5 up- and down-regulated genes for each comparison are indicated. **b** Cell type, and pathway enrichment determined by one-tailed Fisher's exact test in fine brain region-specific gene (for DEG)[18] and OCR (for DAC)[16] sets.

Groups with a low number of significant differential results were masked and colored gray. Neu neuron. pyramidal SS somatosensory pyramidal cells. OPC oligodendrocyte progenitor cells. Odds ratio (OR). "·": Nominally significant (P < 0.05); "+": significant after FDR (Benjamini & Hochberg) correction (FDR < 0.05). Source data is provided as a Source Data file.

More importantly, compared to brain region-specific DEGs, the genes harboring promoter-isoform-specific DEGs have a markedly higher enrichment for common risk variants for multiple neuropsychiatric traits, including SCZ (Fig. 3d). Moreover, the genes harboring promoter-isoform-specific DEGs are strongly enriched for rare coding variants for SCZ and ASD (Fig. 3e). Consistent with this, such genes are strongly overrepresented for synaptic functions (Fig. 3e). Our analysis highlights the alternative promoter usage across different brain regions and their role in neuropsychiatric disorders.

## Brain region-specific enhancer-promoter links at promoter-isoform resolution

Considering the majority of OCRs are distal, we aimed to identify their target genes using the activity-by-contact (ABC) model[38], which combines enhancer activity with the spatial proximity between enhancers and promoters. Recognizing the critical role of promoters in E-P interactions and cis-regulation, and that the 5′ promoter is not necessarily the most active promoter, we refined our analysis to capture enhancer-promoter connections at the

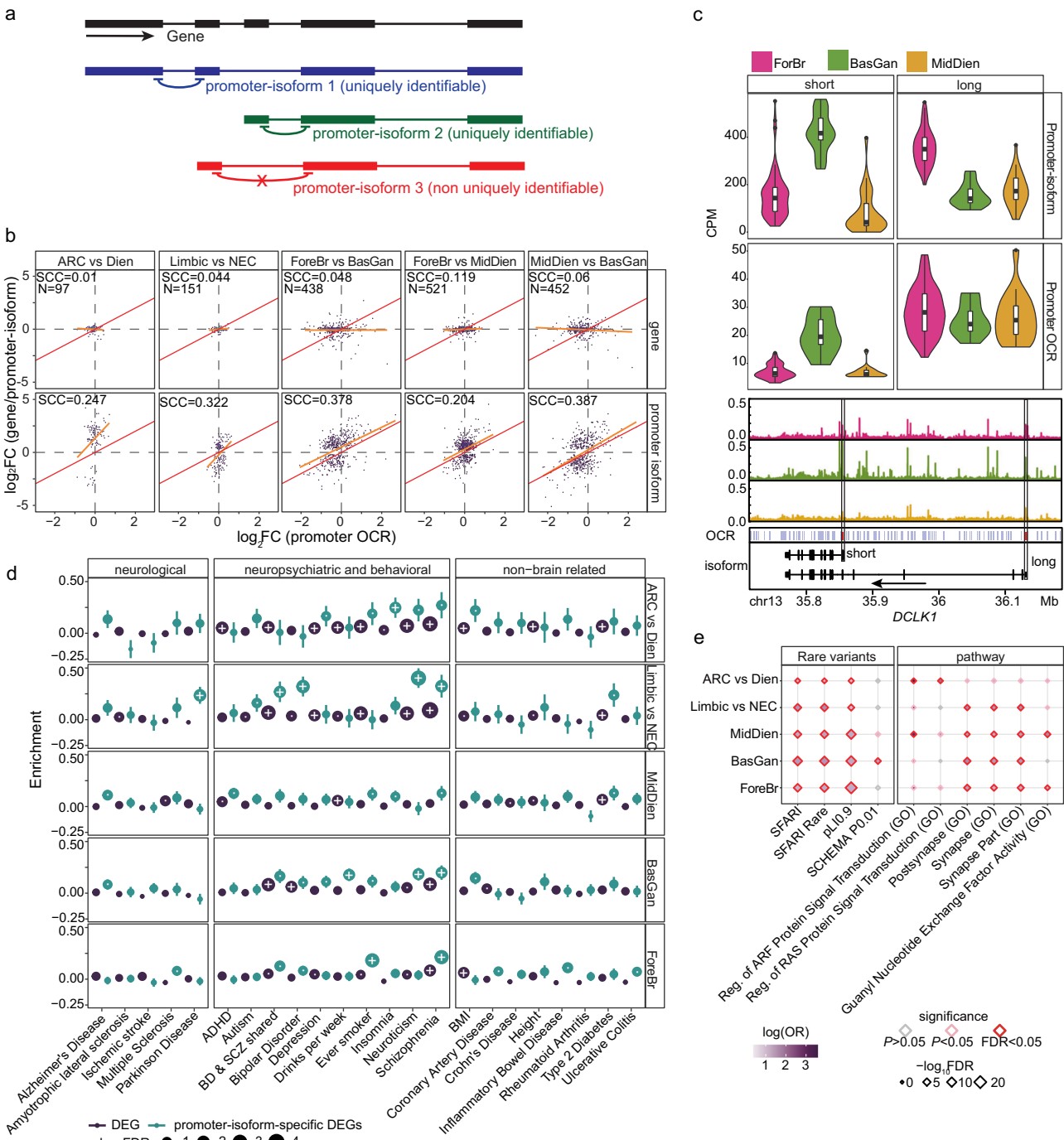

**Fig. 3 | promoter-isoform alternations between brain regions. a** Schematic representation of promoter-isoform quantification using RNA-seq data from neurons. Transcripts that are regulated by the same promoter are grouped (promoter-isoform), and are quantified using the set of unique junction reads spanning the first intron of each transcript. The non-uniquely identifiable promoter-isoforms (red) were removed from this analysis (Methods). **b** Spearman correlation coefficient (SCC) between the log2 fold change of gene/promoter-isoform and promoter OCR across brain regions. $N$ indicates the number of non-concordant promoter-isoforms. Gene promoter OCRs were determined using the 5′ most promoter region. **c** Upper panel, the promoter-isoform expression and promoter OCR chromatin accessibility level across brain regions. For RNA-seq, $N_{ForeBr} = 84$, $N_{BasGan} = 15$, $N_{MidDien} = 27$. For ATAC-seq $N_{ForeBr} = 67$, $N_{BasGan} = 9$, $N_{MidDien} = 17$. Box

plot indicates median, interquartile range (IQR) and 1.5× IQR. Bottom panel, chromatin accessibility profiles (neuronal), and the position of the two promoter-isoforms (highlighted in black boxes). **d** Enrichment of common variants for different trait classes at the DEGs, and promoter-isoform-specific DEGs for each comparison (fine brain region) or each brain region (broad brain region) determined by MAGMA[106]. The y-axis represents the enrichment level (MAGMA beta ± standard error, se), and the size represents the FDR value. "·": Nominally significant ($P < 0.05$); "+": significant after FDR correction (FDR < 0.05). **e** Rare variants, loss of function intolerant genes (pLI0.9), and biological pathways enrichment of the genes that exhibit promoter-isoform-specific DEGs determined by one-tailed Fisher's exact test. FDR corrections were performed using the Benjamini & Hochberg method.

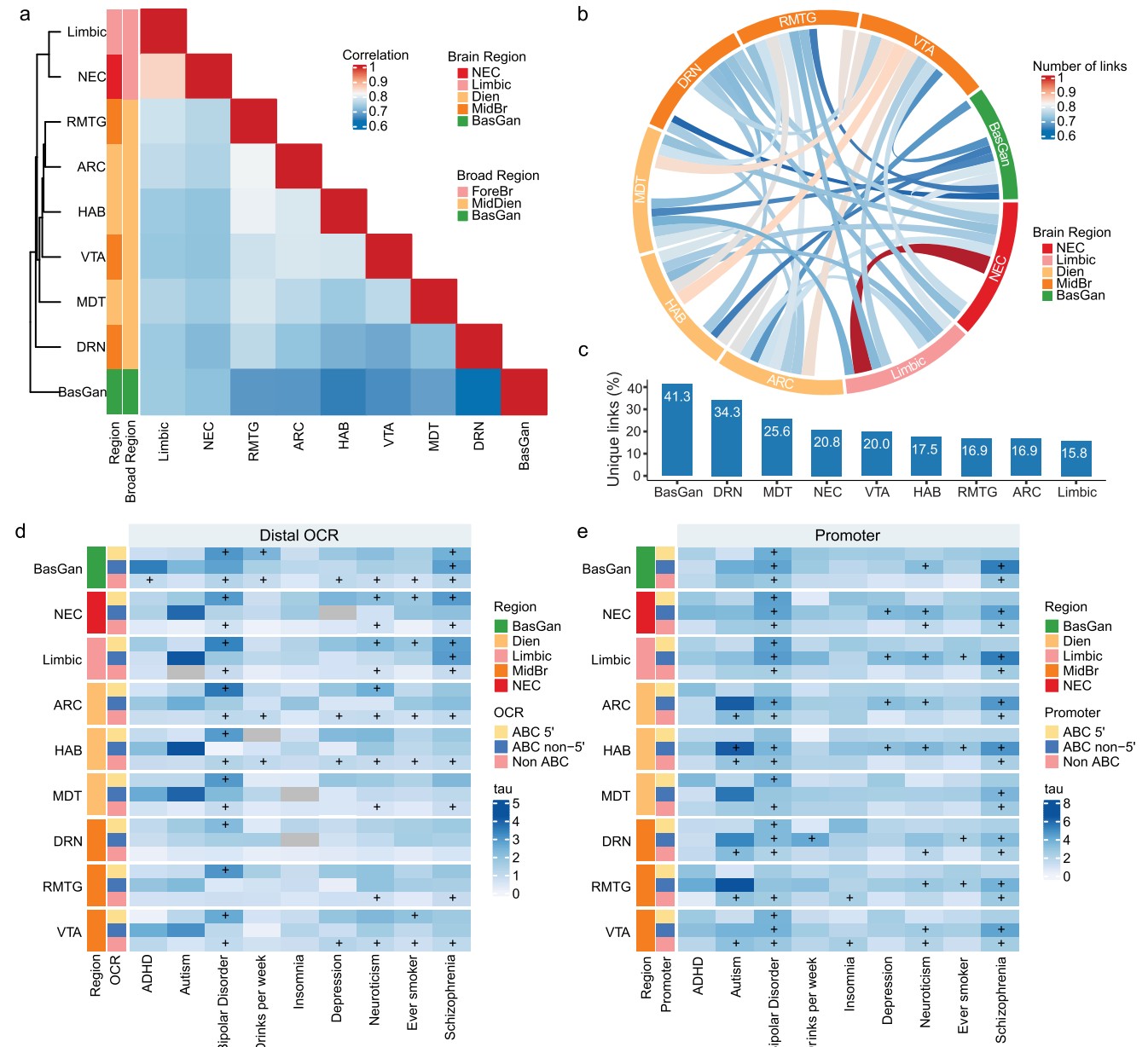

**Fig. 4 | The brain-region specificity of ABC E-P links at isoform resolution.**
**a** Heatmap represents the spearman correlation coefficients of isoform resolution E-P link strength (ABC score) between brain regions. **b** Circos plot of pairwise sharing of E-P links between brain regions. The inner color indicates the number of links. **c** Percentage of unique E-P links for each brain region. Enrichment of common variants for different neuropsychiatric disorders in ABC. **d** Distal OCRs and (**e**), promoters across brain regions. A positive coefficient signifies enrichment in heritability (normalized tau). Negative coefficients were displayed with gray blocks. "+" indicates significant enrichment after FDR (Benjamini & Hochberg) correction (FDR < 0.05). The sidebars indicate brain regions and types of regulatory elements. Source data is provided as a Source Data file.

promoter-isoform level (Methods). Incorporating our brain-region-specific chromatin accessibility profiling and cell-type-specific Hi-C contacts[10,39], we established E-P links at promoter-isoform resolution including both broad brain regions such as BasGan, NEC, and limbic, and fine regions within the MidDien, reflecting their significant molecular diversity.

Across the brain regions, we identified between 24,384 and 30,721 E-P links (ABC > 0.02) at promoter-isoform resolution (Supplementary Fig. 13a, Supplementary Data 10), covering 19,486 to 24,042 enhancer-gene pairs (Supplementary Fig. 13b), with the majorities of the links being within 50 kb (Supplementary Fig. 13c). Notably, 28.78% ± 0.62% (mean ± se, Supplementary Fig. 13d) of genes possessed multiple linked promoter-isoforms, and 38.35% ± 0.45% of the ABC-linked

enhancers were predicted to regulate multiple isoforms (Supplementary Fig. 13e), and 61.20% ± 0.97% of isoforms were regulated by multiple candidate enhancers (Supplementary Fig. 13f). To assess the brain region specificity of E-P links, we compared the pairwise correlation of link strength (ABC score) and the overlaps of links among different brain regions (Fig. 4a, b). As expected, brain regions within the same broad brain regions exhibit higher similarity. While distinct broad brain regions, such as BasGan, displayed a notable 41.3% of unique links (Fig. 4c), consistent with observed gene expression and chromatin accessibility variations across regions (Fig. 1d).

To validate the brain region-specific E-P isoform links, we leveraged independent brain region-specific datasets from the GTEx *cis*-eQTL[40] (Supplementary Fig. 14a), and gene-enhancer coordination

from a multi-brain region single cell atlas[41] (Supplementary Fig. 14b). Our brain region-unique E-P isoform (both 5′ and non-5′) links were corroborated by data from corresponding brain regions or abundant cell types within those regions. For instance, our BasGan unique E-P links were well corroborated by BasGan eQTLs and cis-coordinations of BasGan-restricted MSN (Supplementary Figs. 14 and 15). These results confirmed the reliability of brain region specificity of both 5′ and non-5′ promoter E-P links.

Considering that the majority of neuropsychiatric disorder risk variants are non-coding, we reasoned that these variants would be associated with enhancers and promoters in the brain and exert their effects by disrupting gene regulatory circuits. As such, we assessed the heritability of disease risk attributed to different types of regulatory elements, including 5′ (accounting for 18.9% ± 0.25% of all promoter OCRs) and non-5′ (18.86% ± 0.34%) promoter-isoforms, enhancers that predicted to regulate 5′ (10.97% ± 0.58%) and non-5′ specific (5.02% ± 0.24%) promoters, and non-ABC elements (Fig. 4d, e, Supplementary Fig. 16). As expected, regulatory elements across brain region groups were enriched for disease risk variants, particularly for SCZ and BD (Fig. 4d, e). Notably, ABC model-predicted enhancers demonstrated significantly higher per-single nucleotide polymorphism (SNP) heritability compared to other distal OCRs (Fig. 4d). Furthermore, non-5′ promoter-isoform-specific ABC enhancers showed a similar enrichment to 5′ promoter-isoforms, underscoring the importance of analyzing E-P links with promoter-isoform specificity. Additionally, promoter regions involved in ABC links exhibited substantially higher disease heritability (Fig. 4e), reinforcing the significance of these regulatory connections.

### Fine-mapping of SCZ risk variants

Building on our demonstration of the strong association between genetic risk variants and ABC enhancers of both 5′ and non-5′ promoter-isoforms, we further leveraged these E-P links to characterize the regulatory mechanisms of SCZ risk loci. We focused on fine-mapped risk variants from the latest GWAS study[1] with a posterior inclusion probability (PIP) > 1%, employing the ABC-Max strategy[42] to pinpoint the affected promoter-isoform and genes (Fig. 5a). In total, we prioritized 72 genes across brain regions for 122 out of 4319 fine-mapped SNP overlapped with ABC enhancers (Supplementary Data 11). In line with previous analysis[1], these targets are involved in synaptic vesicle budding and regulation of synaptic plasticity (Supplementary Fig. 17). For validation, we used an orthogonal method, polygenic priority score (PoPS)[43], enhanced by integrating recent brain-related gene features (Methods), which substantially improved the prioritization heritability (Supplementary Fig. 18).

Notably, the majority of genes (60.82% ± 2.56%) were identifiable only through non-5′ specific E-P links (Fig. 5b). And these genes scored significantly higher on PoPS compared to genes linked via 5′ promoters or to the closest genes ($P_{\text{non5' vs 5'}}$ = 2.08e-06; $P_{\text{non5' vs closest genes}}$ = 0.0012, one-tailed Wilcoxon signed-rank test) (Fig. 5c), emphasizing the critical role of promoter-isoform specificity in E-P interactions. For example, the fine-mapped SNP rs7178152 (PIP = 14.53%), was linked to *ABHD2* through a non-5′ promoter-isoform E-P link (Fig. 5d), despite being located closer to the 5′ promoter of *FANCI*. *ABHD2*, implicated in neurotransmitter release[44,45], exhibited a considerably higher PoPS score (0.87, rank percentage) compared to *FANCI* (0.62), and was previously nominated as a causal gene by our enhancer QTL-based fine-mapping study[10].

Furthermore, our analysis highlights the importance of brain region specificity in the genetic regulation of fine-mapped genes, with 21 out of 72 SCZ-prioritized genes being uniquely associated with specific brain regions (Supplementary Fig. 19). For instance, the fine-mapped SNP rs2944829 (PIP = 3.21%) overlapped a BasGan-specific OCR and was linked to *CALN1* through a BasGan specific E-P link (Fig. 5e), even though *CALN1* is expressed across various brain regions (Supplementary Fig. 20a). These brain region-specific

enhancers and their linked target genes provide candidate regulatory mechanisms that may contribute to increased disease susceptibility in certain areas.

### Fine-mapping of BD risk variants

We further applied the promoter-isoform level fine-mapping approach to BD genetic variants (Fig. 6a), focusing on GWAS fine-mapped SNPs (PIP > 1%). Similar to our findings in SCZ, a considerable fraction of BD risk variants was uniquely identified with non-5′ promoter-isoforms (47.61% ± 10.65%) (Fig. 6b). In total, we identified 31 genes linked to 46 fine-mapped SNPs (Supplementary Data 11), which displayed notably high PoPS scores ($P_{\text{non5' vs 5'}}$ = 0.013; $P_{\text{non5' vs closest genes}}$ = 0.037, one-tailed Wilcoxon signed-rank test) (Fig. 6c). For example, the SNP rs1894401 (PIP = 1.11%), linked to the *FURIN* (PoPS = 0.99) known for its role in neuropsychiatric disorders[46,47], is situated near the *FES* promoter (Fig. 6d). Notably, the 5′ promoter-isoform of *FURIN* is not expressed, whereas the 2nd and 4th non-5′ isoforms were highly expressed and linked to the fine-mapped SNP in VTA and NEC, respectively (Supplementary Fig. 20b). In another case, SNP rs7622851 (PIP = 8.38%, Fig. 6e), is connected to the highly expressed non-5′ isoform of the *WDR82* (PoPS = 0.98) gene across multiple MidDien regions, where the 5′ isoform is minimally expressed (Supplementary Fig. 20c). These observations highlight the importance of incorporating promoter-isoform resolution in E-P link analysis.

Extending this analysis to other neuropsychiatric traits (Supplementary Data 11) revealed that 61.51% ± 1.41% of targets were not closest to the lead SNP (Supplementary Fig. 21), 43.91% ± 1.67% were exclusively linked to non-5′ promoter-isoforms (Supplementary Fig. 22), and 27.67% ± 1.04% were involved only in one brain region (Supplementary Fig. 19b). Interestingly, the prioritized non-5′ target genes showed higher PoPS scores compared to their 5′ counterparts (Supplementary Fig. 23), with no significant differences observed in ABC scores (Supplementary Fig. 24) or distances (Supplementary Fig. 25). This comprehensive mapping further underscores the necessity of considering brain region-specific and promoter-isoform-specific regulatory landscapes for a deeper understanding of genetic susceptibility in neuropsychiatric disorders, highlighting the nuanced genetic regulation within these conditions.

## Discussion

Understanding the brain region and cell-type-specific regulome is critical for deciphering the molecular mechanisms of neuropsychiatric disorders. In this study, we present a comprehensive resource of gene expression and chromatin accessibility profiling in neurons and non-neurons across 25 functionally distinct regions of the human brain. In neurons, we identified extensive gene expression and chromatin accessibility alterations across the 4 broad brain regions (ForeBr, MiddDien, BasGan, and HinBr), which are largely driven by the distinct distribution of brain-region-specific neuronal cell subpopulations (Figs. 1f and 2b). Furthermore, our analysis unveiled substantial molecular differences among the fine brain regions, especially within the MidBr and Dien regions (Supplementary Fig. 11a), emphasizing the importance of dissecting molecular diversity and gene regulatory mechanisms in these areas.

In line with recent studies[28,41,48], the NEC brain regions (Supplementary Figs. 8e and 11a), despite their unique cognitive functions, display limited molecular diversity. Among these, the PVC and EC show the most pronounced differences. Intriguingly, a recent study also indicates that, within NEC, the PVC undergoes the most significant alterations in ASD[49]. In addition to the brain-region differences in neuronal cells, we also observed consistent gene expression and chromatin accessibility alterations across brain regions in non-neuronal cells (Fig. 1). However, as non-neuronal brain-region-specific DEGs and DACs were enriched for reference markers of relatively lowly abundant region-specific astrocyte cells (Fig. 1f), our

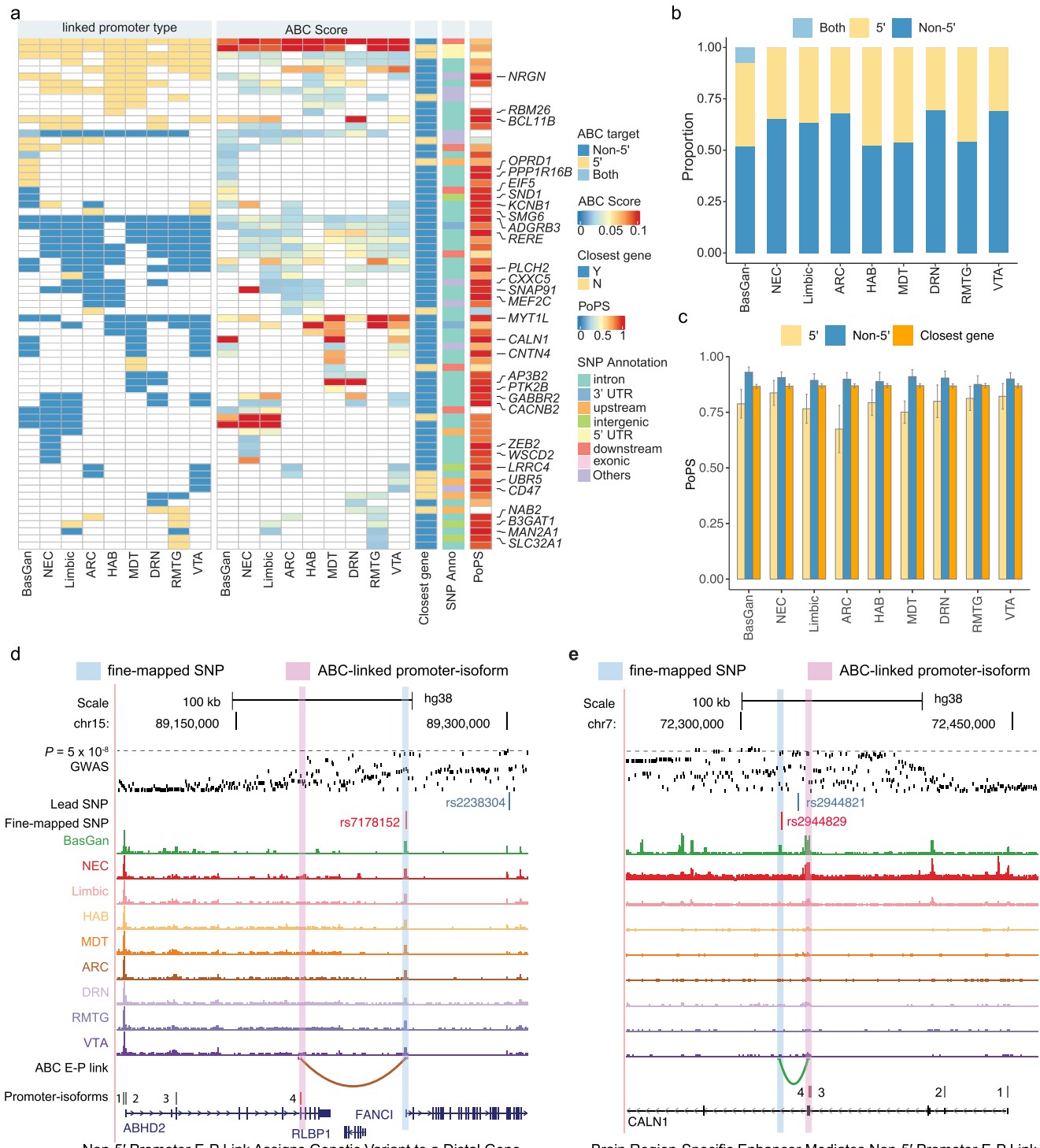

**Fig. 5 | Prioritized genes for schizophrenia fine-mapped SNPs using ABC E-P links at promoter-isoform resolution across brain regions. a** Heatmap showing summary of the E-P link fine-mapped genes for SCZ. The heatmap includes information on the linked promoter type (5', non-5', or both), ABC score, proximity to fine-mapped SNPs (closest gene), SNP annotation, and PoPS score. Genes within the top 5% of PoPS scores are labeled. **b** Proportion of genes identified via 5' and non-5' promoter-isoforms across different brain regions. **c** The distribution of PoPS score (mean ± se) of target genes identified by 5', non-5' isoforms, and genes closest to fine-mapped SNPs. $P_{non5' \, vs \, 5'} = 2.08e\text{-}06$; $P_{non5' \, vs \, closest \, genes} = 0.0012$, one-tailed Wilcoxon signed-rank test. The number of genes can be found in source data. Examples of promoter-isoform E-P link fine-mapped risk variants (**d**), rs7178152, and (**e**), rs2944829. The colors in the E-P link track correspond with the colors of brain regions in the chromatin accessibility track. The numbers in the promoter-isoform track indicate the rank of the promoter-isoforms of target genes (from 5' to 3'), with the fine-mapped promoter-isoforms highlighted in red. Source data is provided as a Source Data file.

ability to determine the heterogeneity of non-neurons was evidently limited.

Having demonstrated the gene expression and chromatin accessibility differences in neurons across brain regions, we next assessed

the regional specificity of *cis*-regulation. We first examined promoter-isoform usage across brain regions and found that, although the majority are consistent with gene expression, a considerable fraction exhibit unique differential expression patterns (i.e., only significant in

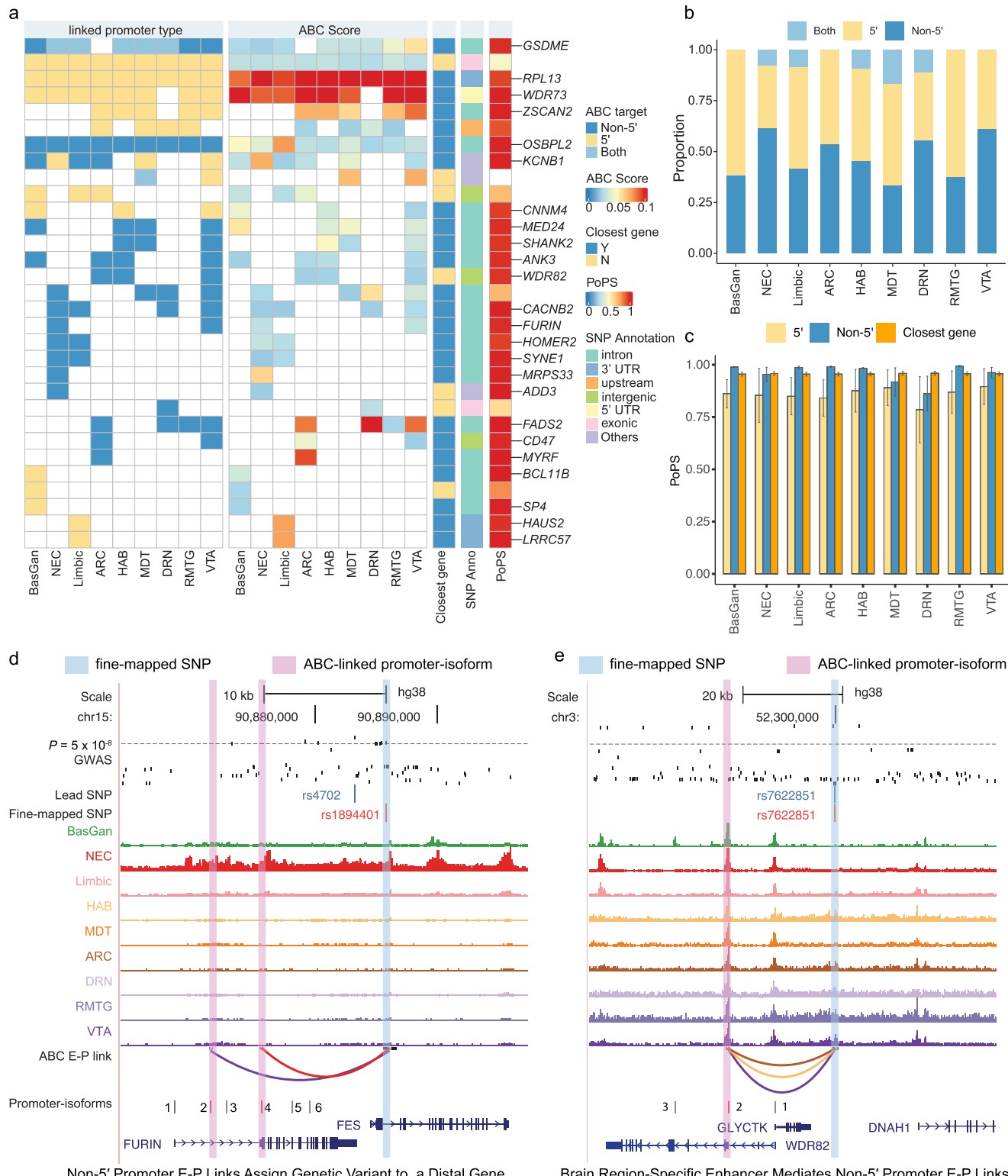

**Fig. 6 | Prioritized genes for bipolar disorder fine-mapped SNPs using ABC E-P links at promoter-isoform resolution across brain regions. a** Heatmap showing summary of the E-P link fine-mapped genes for BD. The heatmap includes information on the linked promoter type (5′, non-5′, or both), ABC score, proximity to fine-mapped SNPs (closest gene), SNP annotation, and PoPS score. Genes within the top 5% of PoPS scores are labeled. **b** Proportion of genes identified via 5′ and non-5′ promoter-isoforms across different brain regions. **c** The distribution of PoPS (mean ± se) score of target genes identified by 5′, non-5′ isoforms, and genes closest

to fine-mapped SNPs. $P_{\text{non5′ vs 5′}} = 0.013$; $P_{\text{non5′ vs closest genes}} = 0.037$, one-tailed Wilcoxon signed-rank test. The number of genes can be found in source data. Examples of promoter-isoform E-P link fine-mapped risk variants (**d**), rs1894401, and (**e**), rs7622851. The colors in the E-P link track correspond with the colors of brain regions in the chromatin accessibility track. The numbers in the promoter-isoform track indicate the rank of the promoter-isoforms of target genes (from 5′ to 3′), with the fine-mapped promoter-isoforms highlighted in red. Source data is provided as a Source Data file.

promoter-isoform, or having a different sign of effect size). The promoter-isoform-specific changes were validated by examining chromatin accessibility of the associated promoter (Fig. 3b), and suggest that the parent genes exhibit alternative promoter usage across brain regions. Interestingly, compared to DEGs, promoter-isoform specific DEGs have a higher enrichment for common disease risk variants, such as for SCZ (Fig. 3d), highlighting the functional role of alternative promoter usage in neuropsychiatric traits. Notably, such genes were strongly associated with synaptic functions.

We then explored enhancer regulation at the level of promoter-isoforms and found that a significant portion of OCRs were linked exclusively to genes through non-5′ promoters. Incorporating promoter-isoform-specific E-P links enhanced the identification of risk genes by connecting SNPs to distal genes rather than the nearest gene. In addition, our results suggest that brain region-specific enhancers can offer insights into the regulatory impact of non-coding SNPs. Importantly, non-5′ promoters often emerge as the major isoforms, with associated risk variants bearing significant biological relevance. A prominent example is the *FURIN* gene, where the 5′ promoter-isoform is not expressed and shows limited chromatin accessibility, while the non-5′ promoter-isoform is highly expressed (Supplementary Fig. 20) and can be linked to fine-mapped risk variants with promoter-isoform specific E-P links. While these promoters do not yield different protein structures[50], they are regulated by distinct genetic programs[46,51,52], underscoring the nuanced regulatory landscapes mediated by different promoter-isoforms.

Overall, our findings highlight the importance of studying gene regulation programs at isoform resolution across different brain regions. While our promoter-isoform resolution genetic fine-mapping successfully identified genes that are missed by gene-level analyses and single brain region approaches, we did not investigate the specific mechanisms by which individual fine-mapped SNPs may influence gene regulation. Future studies are warranted to assess whether these SNPs disrupt the regulatory machinery, such as transcription factor binding motifs, and how they exert their functional effects. As neuropsychiatric traits are closely related to neurodevelopment[53] and, given that the sorted cells examined in this study consist of multiple cell subpopulations, future integration of single-cell ATAC-seq and single cell long read RNA-seq profiling with different developmental-stage, brain regions, and disease status will complement our findings to provide additional insights into the molecular mechanisms of neuropsychiatric disorders.

## Methods

### Description of the post-mortem brain samples
Brain tissue specimens from 25 brain regions (Supplementary Fig. 1) were obtained from 4 donors of European ancestry, 1 Hispanic, and 1 African-American subjects (determined by self-report and ancestry informative marker analysis) with no history of psychiatric disorder, including alcohol or illicit substance abuse (negative toxicology) or were taking neuropsychiatric medications, including benzodiazepines, anticonvulsants, antipsychotics (typical or atypical), antidepressants or lithium. Four subjects (2× males, 2× females) were collected at autopsy at the Brain Endowment Bank at Miller School of Medicine at the University of Miami, and 2 subjects (1× male, 1× female) were collected from Mount Sinai Brain Bank (Supplementary Table 1). Sex was determined by self report, and confirmed with genotype check (Methods and Supplementary Fig. 2). The cause of death, i.e., sudden cardiac death for all 6 subjects, as determined by the forensic pathologist performing the autopsy. All brain specimens were obtained through informed consent and/or brain donation programs at the Miller School of Medicine at the University of Miami and the Icahn School of Medicine at Mount Sinai. All procedures and research protocols were approved by Institutional Review Boards.

### FANS sorting of neuronal and non-neuronal nuclei
Frozen brain samples were homogenized in cold lysis buffer (0.32 M Sucrose, 5 mM CaCl₂, 3 mM magnesium acetate, 0.1 mM, EDTA, 10 mM Tris-HCl, pH8, 1 mM DTT, 0.1% Triton X-100) and filtered through a 40 μm cell strainer. Filtrates were underlaid with sucrose solution (1.8 M Sucrose, 3 mM magnesium acetate, 1 mM DTT, 10 mM Tris-HCl, pH8) and subjected to ultracentrifugation at 107,000 x $g$ for 1 h at 4 °C. Pellets were resuspended in 500 μl DPBS and incubated in BSA (final concentration 0.1%) and anti-NeuN antibody (1:1000, Alexa488 conjugated, Millipore, MAB377X) under rotation for 1 h, at 4 °C. Just prior to FACS sorting, DAPI (Thermo Scientific) was added to a final concentration of 1 μg/ml. Neuronal (NeuN+) and non-neuronal (NeuN-) nuclei were sorted using FACSAria (BD Biosciences).

### Generation of RNA-seq libraries
RNA was isolated from 149 tissue dissections from 25 brain regions. RNA-seq libraries were generated using the SMARTer Stranded Total RNA-seq kit v2 (Takara Cat no. 634419) according to the manufacturer's instructions. Libraries were sequenced with the Illumina Hiseq using 100 bp paired-end reads. After quality controls (see below), we retained 265 RNA-seq libraries, which, on average, corresponds to available RNA-seq data for 22 out of the 25 regions per individual.

### Processing of RNA-seq libraries
Each set of pair-end reads was processed by Trimmomatic[54] to remove low-quality base pairs and sequence adapters. Reads were subsequently aligned to the human reference genome GRCh38 using STAR[55]. To correct for allelic bias resulting from individual-specific genome variation, we ran STAR with the enabled WASP module[56] as we provide both RNA-seq FASTQ file and the Whole Genome sequencing (WGS) file of the corresponding individual. The BAM files that were generated contain the mapped paired-end reads, including those spanning splice junctions. Following read alignment, expression quantification was performed at the transcript isoform level using RSEM[57] and then summarized at the gene level. Gene quantifications correspond to GENCODE[58]. Quality control metrics (Supplementary Data 2) were reported with RNA-SeqQC[59], Qualimap[60], and Picard.

### Generation of ATAC-seq libraries
ATAC-seq reactions were performed on Neuronal (NeuN+) and non-neuronal (NeuN-) nuclei isolated by FANS from 149 tissue dissections from 25 brain regions, resulting in 291 ATAC-seq libraries. Where available, 100,000 sorted nuclei were centrifuged at 500 × $g$ for 10 min, 4 °C. Pellets were resuspended in the transposase reaction mix and libraries were generated using an established protocol[61]. Libraries were sequenced with the Illumina Hiseq 4000 using 50 bp paired-end reads. After quality controls, we retained 210 ATAC-seq libraries, which, on average, corresponds to available ATAC-seq data for about 18 out of the 25 regions per individual.

### Generation of whole-genome sequencing libraries
Whole-genome sequencing libraries were prepared at BGI Genomics. DNA was extracted from frozen tissue sections using the QIAamp DNA mini kit (Qiagen, Cat no. 51306) according to the manufacturer's instructions. Whole-genome data were generated for all 6 subjects on Illumina Hiseq 4000 using 100 bp paired-end reads.

### Data processing and quality control of whole-genome sequencing libraries
To facilitate the alignment of raw sequencing files and perform variant calling, we utilized CCGD pipeline (https://github.com/CCDG/Pipeline-Standardization/)[62]. In brief, reads were aligned to hg38 human reference genome using BWA-MEM. Then, the pipeline follows GATK Best Practises guidelines to perform duplicate marking, base recalibration, indel-realignment, quality score binning, and variant calling. All

samples were noted to have broadly even profiles across quality control metrics, i.e genome coverage >98% (10x coverage > 96% loci), dbSNP coverage >98%, and transition/transversion (Ti/Tv) ratios between 2.07 and 2.08, consistently with our genome-wide expectations (general range for known and novel loci on the human genome is ~2.0−2.1[63], while the empirical value for Illumina Hiseq platform is 2.07[64]) (Supplementary Data 1).

To verify ancestry information, we merged the whole genome samples with 1KG cohort (http://ftp.1000genomes.ebi.ac.uk/vol1/ftp/release/20130502/supporting/GRCh38_positions/) and performed PCA on the thinned set of 30,000 randomly selected SNPs with MAF ≥ 5% (SNPRelate package v1.16[65]). Based on the proximity of our samples to 1KG population clusters in the two-dimensional space of the first two principal components, we checked the ethnicity information of all six individuals (Supplementary Fig. 2a).

### Quality control of RNA-seq libraries

On average, over 51 million sequenced paired-end reads were obtained for each sample. In our initial dataset of 308 samples, 17 samples had a technical or biological replicate. We decided to keep those replicates with a better correlation to the gene expression profiles of the other samples originating from the same cell type and brain region. In case of similar results, we retained a sample with a higher number of uniquely mapped reads. Then, we calculated the correlation of each sample to all other samples of the same cell type and highlighted 26 samples with markedly different correlations, i.e., the difference in mean correlation of the given sample with the rest of the dataset and the mean correlation of all pairs of samples within the dataset was more than twice higher than the standard deviation calculated upon all pair's correlation. All those 26 samples originating from 15 distinct brain regions of 5 individuals were removed after a visual inspection in IGV combined with an inspection of metadata which identified probable reasons such as the low amount of starting material, low RIN, or low ratio of uniquely mapped reads. When we applied this filtering procedure, we ended up with a final set of 265 samples, i.e., 132 neuronal and 133 non-neuronal (Supplementary Data 2) that are well separated in PCA (Supplementary Fig. 8d). To check the sex of individuals within our cohort, we also measured the number of reads mapped on genes located on chromosome Y (genes located on pseudoautosomal regions are not counted; Supplementary Fig. 2b). To check the identity of RNA-seq samples, we ran genotype comparisons of all samples against each other and against imputed genotypes from WGS (Supplementary Fig. 2d).

### Quality control of ATAC-seq libraries

On average, over 57 million sequenced paired-end reads were obtained for each sample. Because of using FANS sorted nuclei as opposed to whole cells, only a low fraction of the reads were mapped to the mitochondrial genome (mean of 0.97% of the uniquely mapped reads). In our initial dataset of 293 libraries, seven libraries had a technical or biological replicate. We decided to keep those replicates with a better correlation to the chromatin accessibility profiles of the other samples originating from the same cell type and brain region. We also excluded libraries that had low mappability (less than 50%), low per-sample called OCRs (less than 3000), low GC content (less than 90% of cell type median, i.e., 52.15% and 54.35% for neuronal and non-neuronal libraries, respectively) or low final read count (less than 5,000,000). The threshold of cell GC content was set empirically by testing all values between 75 and 95% of cell-type median (with a step of 5%) and observing changes in the clustering analysis as those samples with low GC were frequently outliers in the MDS plots. Additionally, we inspected all ATAC-seq libraries in IGV browser and removed an additional 8 samples with the lowest TSS enrichment and/or less than 1000 nuclei. When we applied this filtering procedure, we ended up with a final set of 202 samples, i.e., 97 neuronal and 105 non-neuronal

(Supplementary Data 3). Those neuronal and non-neuronal samples are relatively well separated in PCA (Supplementary Fig. 8d). Similarly to RNA-seq, we performed a sex check (Supplementary Fig. 2c) and a genotype check (Supplementary Fig. 2e). ATAC-seq QC metrics are summarized in Supplementary Data 3.

For further steps, we split samples into neuronal and non-neuronal datasets. The rationale for this decision comes from the differential analysis that was unable to properly correct for the effect of markedly different chromatin compositions of these cell types.

### Genetic concordance analysis

To verify the identity of samples across all assays, we compared called genotypes of RNA-seq and ATAC-seq samples against whole-genome sequencing samples using KING v1.9[66]. To overcome the issue of a relatively high error rate for variant calling in functional genomics assays, we utilized GATK Best Practises guideline (https://software.broadinstitute.org/gatk/best-practices/workflow?id=11164), followed by the removal of variants with minor allele frequencies (MAF) < 25%. For the RNA-seq cohort, this analysis resulted in the correction of genotypes of 2 unambiguously swapped samples and the removal of 2 samples due to genotype contamination, i.e., high genetic concordance of a single sample with multiple distinct genotypes. In the case of ATAC-seq, we corrected the genotypes of 4 unambiguously swapped samples and removed 5 likely contaminated samples.

### Processing of ATAC-seq libraries

The raw reads were trimmed with Trimmomatic[54] and then mapped to the human reference genome GRCh38 analysis set reference genome with the pseudoautosomal region masked on chromosome Y with the STAR aligner[55]. To correct for allelic bias resulting from individual-specific genome variation, we ran STAR with enabled WASP module[56] as we provide both ATAC-seq FASTQ file and WGS file of the corresponding individual. This yielded for each sample a BAM file of mapped paired-end reads sorted by genomic coordinates. From these files, reads that mapped to multiple loci or to the mitochondrial genome were removed using samtools[67] and duplicated reads were removed with Picard. Quality control metrics (Supplementary Data 3) were reported with phantompeakqualtools[68] and Picard.

Reads from the same brain region and cell type were subsequently subsampled and merged, creating 50 BAM files with a uniform depth of 170 million paired-end reads. For neuronal samples from the GP, HAB, VTA, and DRN, we had less than 170 million pair-end reads (39, 123, 136, and 162 million, respectively), so retained all reads from the corresponding samples. With the exception of those samples, all subsampling ratios were calculated per each sample individually within the 50 respective groups (brain region by cell type) to ensure that each contributes the same number of reads regardless of their overall read counts. bigWig files were created using these BAM files and peaks were called with MACS2[69]. After removing peaks overlapping ENCODE blacklisted regions of anomalous, unstructured, or high signal in functional genomics assays[70], 320,308 and 196,467 peaks remained for neuronal and non-neuronal datasets, respectively. For each peak, we assigned the closest gene and the genomic context of an ATAC-seq OCR using ChIPSeeker[71]; the transcript database was built by GenomicFeatures[72] upon ENSEMBL genes. Finally, read counts of all samples were quantified within these peaks using the featureCounts function in RSubread[73].

### Analysis of differentially expressed/accessible genes, promoter-isoforms, and OCRs

To assess which genes, isoforms, and OCRs showed differential expression and accessibility, we employed statistical modeling based on linear mixed models. The starting point here was three count matrices with raw read counts per each sample and feature (i.e., gene, promoter-isoform, or OCR). For gene expression and chromatin

accessibility, we excluded features that were lowly expressed/accessible by only keeping those with at least 1 count per million (CPM) reads in at least 20% of the samples. For promoter-isoforms, we used a more stringent threshold, and only kept those with at least 2 CPM reads in >40% of the samples. Then, the read counts were normalized using the trimmed mean of M-values (TMM) method[74]. Covariates were selected as implemented in our previous study[8] (Supplementary Data 4). Statistical Analysis of differences in gene/isoform expression and chromatin accessibility: The normalized read count matrices from voomWithDreamWeights (variancePartition package[75]) was then modeled by fitting weighted least-squares linear regression models estimating the effect of the right-hand side variables on the expression/accessibility of each feature.

As our dataset contains up to 25 samples per individual within both neuronal and non-neuronal subsets, we ran differential analysis by dream[76] method from variancePartition package[75]. Dream properly models correlation structure and, thus, keeps the false discovery rate lower than the other commonly used methods for this purpose. Finally, adjusted matrices of gene/promoter-isoform expression and chromatin accessibility were created for neuronal and non-neuronal samples where the effects of Sex and technical covariates were removed.

## Principal component analysis
After regressing the selected covariates, we applied Singular Value Decomposition (SVD) to the residual matrices using R's svd function. The first two columns of the v matrix were used as PC1 and PC2. The percentage of variations was determined with $\frac{d_i^2}{\Sigma d^2}$.

## Canonical correlation analysis (CCA)
CCA between gene expression and chromatin accessibility was performed based on R toolkit Seurat[77] and Signac[78]. Briefly, we determined gene activity score for all the expressed genes from chromatin accessibility across each sample. We focused on the protein-coding genes that are highly variable for gene expression (top 2000 dispersion, aka variance to mean ratio) or gene activity (top 2000 dispersion). Then we utilized a variant of CCA, diagonal CCA[26] to construct our canonical correlation vectors.

## Covariates selection and differential analysis
The starting point for statistical modeling of gene/isoform expression and chromatin accessibility was chosen with the variables Brain_region (25 levels) and Sex (2 levels) for a base model. Sex was included as it is known to have a strong effect on a few genes/promoter-isoforms/OCRs (features) primarily located on the sex chromosomes. To assess which covariates should be included in order to have a good average model for gene/isoform/OCR accessibility, we employed the Bayesian information criterion (BIC) as implemented in our previous study[8]. This procedure pinpoints the best-performing covariates upon the initial sets of 79 and 51 covariates for RNA-seq and ATAC-seq datasets, respectively. We required to net improve at least 5% of the features showed a change of 4 in the BIC score, which is above the lower boundary of "positive" evidence against the null hypothesis. Summaries of selected covariates for all combinations of analysis type and cell type are provided in Supplementary Data 4.

## Promoter-isoform analysis
We employed the proActiv package to determine the promoter-isoform expression[35]. Briefly, we first identified the uniquely identified promoter-isoform (Supplementary Data 7), which excludes the isoform that is a single exon. We assign the isoform-transcript id as the promoter-isoform; when a promoter-isoform corresponds to multiple isoforms, we randomly choose one. Then we quantified the promoter-isoform expression by estimating the junction reads aligned to the first intron. We used the same differential analysis for promoter-isoform analysis

(Supplementary Data 8) and excluded the internal promoter (first intron edge of current isoform overlapped with non-first intron edge of other isoforms, Fig. 3a) as it's challenging to accurately quantify the expression level[35]. To determine the promoter-isoform-specific DEGs, we focused on the significant (FDR < 0.05) promoter-isoforms between broad region pairwise comparisons. We selected such promoter-isoforms that the genes are not significant (nominal $P > 0.1$) or have an opposite fold change direction (Supplementary Data 9).

## Activity-by-contact (ABC) gene-enhancer links at promoter-isoform resolution
To identify active promoters, we also included internal promoters, requiring both detectable promoter-isoform expression (CPM > 1 in at least 20% of samples) and adjacent chromatin accessibility (OCR peaks within 1 kb of the transcription start site (TSS)). Due to pronounced molecular diversity within MidDien brain regions, our analysis focused on neuronal cells across brain region groups: NEC, limbic, BasGan, and all fine regions within MidDien. By combining brain region-specific promoter-isoform annotations and chromatin accessibility with cell-type-specific Hi-C data[10,39], we established promoter-isoform level E-P links using the ABC model[38]. In accordance with the authors' directions, we filtered out predictions for genes on chromosome Y and lowly expressed genes (genes that did not meet inclusion criteria in our RNA-seq dataset). We used the default threshold of ABC score (a minimum score of 0.02) and the default screening window (5MB around the TSS of each gene).

The pair-wise correlation of E-P link strength among brain regions was evaluated with Spearman correlation using complete values (i.e., overlapped links). The correlation heatmap was visualized with R package ComplexHeatmap (v2.6.2). The hierarchical clustering was performed with Euclidean distance and default parameters.

## Validation of brain region-unique gene-enhancer links
We validated the brain region unique-E-P links using two independent datasets, GTEx cis-eQTLs from matched brain regions and gene-enhancer correlation from a multi-brain region single cell atlas (BICCN)[41]. For GTEx cis-eQTLs, we intersected ABC enhancers (flanking 500 bp for both up- and down-stream) with the genomic coordinates of cis-eQTLs and SNPs within the same LD block of the lead SNP ($r^2 > 0.8$). For BICCN gene-enhancer correlated data, we intersected the ABC enhancer regions with BICCN enhancers. Then, we evaluated the validated proportion by dividing the number of validated enhancer-targets by the number of ABC links with intersected enhancers. A one-tailed Fisher's exact test was applied to examine whether the validated proportion for one region is higher than other regions. We evaluated 5' and non-5'-links separately, which were grouped by the types of promoter-isoforms.

## Fine-mapped and lead SNPs for neuropsychiatric disorders
For SCZ, we utilized the findings from the GWAS conducted by ref. 1, where FINEMAP[79] was applied for statistical fine-mapping to nominate potential causal variants. For BD, we performed statistical fine-mapping using LD matrices derived from individuals of European ancestry from the UK Biobank, alongside GWAS summary statistics[2], in accordance with the methodologies outlined by Koromina et al.[80]. In this process, we focused on prioritizing target genes for variants exhibiting a PIP > 1%. Additionally, we identified lead SNPs from the GWAS analysis of neuropsychiatric disorders. Here, we flanked significant variants on both sides by 10 kb, collapsed these windows with overlaps, and obtained the lead SNP with the lowest $P$ value for each window (Supplementary Data 11).

The gene-based annotation of variants was evaluated with ANNOVAR[81] (v2020-06-08) using dbNFSP version 3.0a (hg38). The closest genes to variants were identified with the closest-features program in BEDOPS[82] (V2.4.41).

## Partitioned heritability with stratified LD score regression

We partitioned heritability for DE peaks/TEns as well as top eSNPs to examine the enrichment of common variants in neuropsychiatric traits with stratified LD score regression (v.1.0.1)[83] from a selection of GWAS studies, including:

Attention-deficit/hyperactivity disorder (ADHD)[84], Autism spectrum disorder[85], Bipolar Disorder and Schizophrenia shared (BD & SCZ shared)[86], Bipolar Disorder (BD)[2], Major depression[87], Drinks per week[88], Ever smoker[88], Insomnia[89], neuroticism[90], Schizophrenia (SCZ)[1], Alzheimer's Disease[91], Amyotrophic lateral sclerosis[92], Multiple Sclerosis[93], Parkinson Disease[94], body mass index (BMI)[95], Coronary Artery Disease[96], Crohn's Disease[97], Height[98], Inflammatory Bowel Disease[97], Ischemic stroke[99], Rheumatoid Arthritis[100], Type 2 Diabetes[101], Ulcerative Colitis[97].

Briefly, with the input peaks, a binary annotation was created by marking all HapMap3 SNPs[102] that fell within the peak or eSNPs and outside the MHC regions. LD scores were calculated for the overlapped SNPs using a LD window of 1 cM using 1000 Genomes European Phase LD reference panel[103]. The enrichment was determined against the baseline model[83]. We also determined the gene sets enrichment with the same pipeline (Fig. 4d, e). The genes were padded by 35 kb upstream and 10 kb downstream, and HapMap3 SNPs that fell within such regions and outside of the MHC regions were utilized. To enable comparisons across traits, we utilized the regression coefficient (normalized tau), and its *P* value (hypothesis, tau >0). The normalized tau measures the average per-SNP contribution of the annotation to heritability.

## Overlap of OCR with existing annotation

We collected cell-type-specific cross-brain region OCRs from a recent single-cell ATAC-seq reference[16]. First, we compared the Jaccard index between our OCRs with the reference and confirmed the cell type and brain region specificity. In addition, we utilized the union of all peaks as background and performed a single-side Fisher's exact test between the DAC and the single-cell makers.

## Gene set enrichment analysis

To explore the function of a gene set, we collected functional gene sets from MSigDB 7.0[104], and human brain single-cell markers[18,28] (For Skene et al., top 1 specificity percentile were used) One-tailed Fisher's exact tests were used to test the enrichment and significance. In addition, we have performed the gene set enrichment with SynGO[105] with default parameters. In such analysis, we used all expressed genes relevant to the specific group as background.

To examine the genetic enrichment of gene sets, we used MAGMA (v 1.07b)[106] with GWAS data (described above). Briefly, protein-coding genes were padded by 35 kb upstream and 10 kb downstream, and the MHC region was removed due to its extensive linkage disequilibrium and complex haplotypes. The European panels from 1000 Genome Project phase 3 were used to estimate the Linkage disequilibrium (LD)[103]. The BETA value from the MAGMA output was used to represent the enrichment. To determine the pathways that are enriched for SCZ common variants, we determined the MAGMA enrichment for all the above-mentioned MSigDB pathways and collected all the significant pathways after FDR (Benjamini & Hochberg) correction (FDR < 0.05)[107].

## Polygenic priority score gene prioritization

We used PoPS[43] to validate our findings and enhanced PoPS gene prioritization by incorporating evolutionary[108] and mutational constraint[109] data, recent single-cell/nucleus profiling of the human brain[48,110–112], as well as co-expression networks in control[113] and psychiatric disorder[114] contexts. Our analysis was restricted to 18,383 protein-coding genes. Single-cell/nucleus data were processed using Scanpy[115], generating several key features: (1) Dimensionality Reduction: Post-normalization and identification of highly variable genes, PCA was performed using truncated SVD and Independent components analysis (ICA) with FastICA[116]. The derived PCs and ICs were then projected to the entire gene set. (2) Expression Metrics: We calculated the average gene expression at pseudobulk levels for various cell annotations. To mitigate variance in low-expression genes, we normalized the count matrix using voom[117], followed by normalization for cell-type specificity of gene expression[18]. (3) Differential Expression Analysis: Utilizing Pegasus[118], we conducted differential expression analysis across cell annotation hierarchies, focusing on intra-cell type comparisons to identify subtype-specific expression patterns. (4) Gene Expression Programs: Non-negative Matrix Factorization[119] was applied to decipher gene expression programs. For co-expression networks, we incorporated module assignments and module membership (kME) values. All features were standardized. We incorporated the additional features with the features reported from the original method[43] to determine PoPS scores for different neuropsychiatric traits. To assess the added features, we performed S-LDSC to assess the per-SNP heritability of the prioritized genes (top 10%) and found the inclusion of new features substantially increased per-SNP heritability across various neuropsychiatric traits (Supplementary Fig. 18).

## Reporting summary

Further information on research design is available in the Nature Portfolio Reporting Summary linked to this article.

## Data availability

The raw data generated in the current study are available through the Gene Expression Omnibus (GEO) under accession number GSE211826 (ATAC-seq and RNA-seq), and Sequence Read Archive (SRA) under accession number PRJNA870417 (WGS). The UCSC genome browser tracks of our processed ATAC-seq data, download links, and the updated PoPS score feature and PoPS score for different traits are available at our webpage at synapse (synID:syn35856920). The following publicly available datasets were used: The Corces et al. human multi-brain-region single-cell ATAC-seq reference[16] is available on NCBI GEO (GSE147672), the Skene et al. mouse multi-brain-region single-cell RNA-seq reference[18] is available from http://www.hjerling-leffler-lab.org/data/scz_singlecell/, the Lake et al. human brain single-cell RNA-seq reference[120] is available on NCBI GEO (GSE97942), GTEx human multi-brain region RNA-seq[40] data is available on from https://gtexportal.org/home/, the Fullard et al. human multi-brain region ATAC-seq data[8] is available on NCBI GEO (GSE96949). Source data are provided with this paper.

## Code availability

The code is available at Zenodo https://zenodo.org/records/13922059 (Ref. 121).

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

## Acknowledgements

We thank the computational resources and staff expertise provided by the Scientific Computing of the Icahn School of Medicine at Mount Sinai. This study was supported by the National Institute of Mental Health: NIH grants nos. R01-MH109677 (P.R.), U01-MH116442 (P.R. and V.H.), R01-MH125246 (P.R.) and RF1-MH128970 (P.R.), and the National Institute on Aging: NIH grants nos. R01-AG050986 (P.R.), R01-AG067025 (P.R. and V.H.) and R01-AG065582 (P.R. and V.H.). P.D. was supported in part by NARSAD Young Investigator Grant 29683 from the Brain & Behavior Research Foundation. G.E.H. was supported in part by NARSAD Young Investigator Grant 26313 from the Brain & Behavior Research Foundation. J.B. was supported in part by Alzheimer's Association Research Fellowship AARF-21-722200. Figure1a created using BioRender (https://biorender.com).

## Author contributions

P.R. conceived of and designed the project. J.F.F. and P.R. designed experimental strategies for epigenome profiling of human postmortem tissue. D.A.D., V.H., and W.K.S. dissected and provided brain specimens. R.M. prepared nuclei and performed FANS. R.M., Z.S., and J.E. generated ATAC-seq and RNA-seq libraries. S.A. and N.L. performed sequencing of ATAC-seq and RNA-seq libraries. P.D. and P.R. designed analytical strategies. J.B. and P.D. conducted initial bioinformatics, sample processing and quality control. P.D. and L.S. performed the downstream analysis. J.F.F and P.R. supervised data generation. G.E.H. and P.R. supervised data analysis. P.D., L.S., and P.R. wrote the manuscript with input from all authors.

## Competing interests

Boehringer Ingelheim Pharma GmbH & Co. KG supported this work only by providing financial support. S.A. and N.L. are employees of Boehringer Ingelheim Pharma GmbH & Co. KG. Aside from financial support, the industrial sponsors had no role in the design of this study, the sample collection, analysis, or interpretation of data, the writing of the report, or in the decision to submit the article for publication. The remaining authors declare no conflicts of interest.

## Additional information

[1]Center for Disease Neurogenomics, Icahn School of Medicine at Mount Sinai, New York, NY, USA. [2]Friedman Brain Institute, Icahn School of Medicine at Mount Sinai, New York, NY, USA. [3]Department of Psychiatry, Icahn School of Medicine at Mount Sinai, New York, NY, USA. [4]Department of Genetics and Genomic Science, Icahn School of Medicine at Mount Sinai, New York, NY, USA. [5]Brain Endowment Bank, Department of Neurology, Miller School of Medicine, University of Miami, Miami, FL, USA. [6]Department of Neuroscience, Icahn School of Medicine at Mount Sinai, New York, NY, USA. [7]Mental Illness Research Education and Clinical Center (MIRECC), James J. Peters VA Medical Center, Bronx, NY, USA. [8]John P. Hussman Institute for Human Genomics and Dr. John T. Macdonald Foundation Department of Human Genetics, Miller School of Medicine, University of Miami, Miami, FL, USA. [9]Global Computational Biology and Digital Sciences, Boehringer Ingelheim Pharma GmbH and Co. KG, Biberach, Germany. [10]Center for Precision Medicine and Translational Therapeutics, James J. Peters VA Medical Center, Bronx, NY, USA. ✉e-mail: pengfei.dong@mssm.edu; panagiotis.roussos@mssm.edu

