## [Transparent Peer Review file · Nature Communications]

A Multi-Regional Human Brain Atlas of Chromatin Accessibility and Gene Expression Facilitates Promoter-Isoform Resolution Genetic Fine-Mapping

Corresponding Author: Dr Panos Roussos

Version 1:

Reviewer comments:

Reviewer #1

(Remarks to the Author)

Dong et al. present a heavily revised manuscript that now focuses on the mapping of different promoter isoforms and the associated gene regulatory interactions that they identify by combining ATAC-seq and RNA-seq data across a large compendium of bulk datasets acquired from a diversity of brain regions. I commend the authors for such a large revision. The amount of work represented in this revision is substantial and I feel that the manuscript is much improved by the new inclusions and omissions and the more concrete focus on promoter-isoforms. Many of my initial concerns have been addressed, either by inclusion of updates or omission of analyses so I focus my comments on a small number of concerns that came up in the modified manuscript.

The authors make a large distinction between 5' and non-5' promoter isoforms. I acknowledge that a distinction is needed but I'm not familiar with why the 5' vs non-5' distinction is used. I would find it more helpful to use a "default" vs "non-default" distinction, for example by using the promoter annotated by RefSeq vs alternative promoters. The reason I mention this is that in the current distinction, if the authors find that a non-5' promoter is used, I'm not sure how to assess that because if that non-5' promoter is still the default promoter, then the authors results are less novel than they would be if the non-5' promoter was something else / something new. If my interpretation of this is mis-informed and in fact the 5' isoform is actually the default isoform, then this concern is not valid; however, that is not my understanding.

While this is not my specific expertise, I am not used to PIP being presented as a percentage but rather a fraction. And in that capacity, I am used to seeing PIP scores greater than 0.9 for likely causative fine-mapped variants. However, the examples that the authors use have PIP scores less than 15%. Does this correspond to a PIP less than 0.15 or is this metric somehow inverted and it represents a PIP greater than 0.85? Can the authors clarify this discrepancy and (if relevant) justify why their PIP scores appear to be so low? It would certainly seem outside of the standard in the field to be highlighting variants with a PIP less than 0.15. Again, not my expertise but this seems like a nonstandard presentation and it might behoove the authors to use the more standard presentation of PIP.

For the fine-mapped examples presented in Figs. 5d-e and 6d-e, could the authors provide additional layers of information that might lend more confidence to these being the causal SNPs? For example, if these variants are thought to be acting by perturbing gene regulation, do they reside within a transcription factor motif and is the base change something that would be damaging to TF binding?

Related to 1.2 Response, I think it would be appropriate for the authors to modify how they discuss the number of samples obtained in the study to accurately capture the difference between the total number of samples assayed (25 brain regions across 6 individuals) and the actual number of pass-filter samples. However, I know that the authors have already made their preference clear on this matter and they have also removed the number of brain regions from the title so I leave this suggestion to editorial discretion.

Reviewer #3

(Remarks to the Author)

The authors base their response to questions of novelty and significance on the observed "alternative promoter usage" but provide no response to the question of functional validation. In the absence of orthogonal validation, I worry that the

observation stands without biological meaning.

Response to the reviewer

Reviewer #1:

Remarks to the Author:

Dong et al. present a multi-region atlas of brain chromatin accessibility and gene expression from 6 individuals. They use this atlas to characterize genetic risk variants for neuropsychiatric traits, eventually focusing on a single promoter-based module that they argue represents a novel association with the disease. Overall, I don't find the approach or overall conclusions to be particularly novel and feel that many of the results presented have been largely described previously. I have significant reservations about the quality of some of the data sets generated, particularly the ATAC-seq data. The primary novelty of the paper comes from the promoter-isoform analyses, which are at times confusing, and the pr_7 module, though I feel that some of the conclusions drawn from the pr_7 analyses are misguided or over-interpreted. I've outlined more specific comments below, divided into major suggestions and minor suggestions.

1 Response

We are grateful for the reviewer's thorough evaluation and constructive feedback. In response to the concerns raised, we have undertaken significant revisions to enhance the clarity, quality, and novelty of our manuscript.

To address the concerns regarding the quality of our datasets, specifically the ATAC-seq data, we conducted a comprehensive comparison with similar postmortem studies recently published in the field. This comparison assured us that our RNA-seq and ATAC-seq data are of comparable quality. Additionally, we have taken the proactive step of excluding 8 samples identified as being of lower quality to ensure the integrity and reliability of our findings.

Recognizing the reviewer's reservations about the co-expression/accessibility analysis, we have elected to remove this portion of our work from the manuscript. This decision was made to focus on the most robust and innovative aspects of our research.

We are particularly thankful for the reviewer's recognition of the novelty in our promoter-isoform analyses. In revising our manuscript, we have further elaborated on our development of enhancer-promoter links at the promoter-isoform resolution. Our findings reveal that distal open chromatin regions (OCRs) associated with non-5' promoters exhibit significant heritability for neuropsychiatric disorders on a per-SNP basis. This nuanced approach has uncovered critical links between risk variants and target genes, offering new insights into the regulatory mechanisms implicated in neuropsychiatric disorders and highlighting the unique contribution of our study to understanding the genetic and molecular underpinnings of these conditions.

We believe these revisions and clarifications address the reviewer's concerns and significantly enhance the manuscript's contribution to the field.

Major Suggestions

1. The quality control metrics used for ATAC-seq are not consistent with what has become the "gold standard" in the field (e.g. ENCODE - <https://www.encodeproject.org/atac-seq/>). The primary metrics used (as per Sup. Note) are mappability, number of peaks called, GC content, and final read count. These metrics do not measure

the actual quality of the data which could otherwise be described as the signal-to-noise ratio. This is typically calculated as a **TSS enrichment score** in ATAC-seq. The authors do present the fraction of reads in promoters, a similar metric, in Sup Table 4 but it varies widely from as low as 4% to as high as 82%. I'm surprised to see both of these extremes – certainly a sample with 4% of reads in promoters is not of passable quality and similarly a sample that has 82% of reads in promoters would also worry me as this data might be unreasonably biased. On top of this, I am perplexed as to how this metric is calculated as some samples have 82% of reads in promoters but only 10% of reads in peaks. This seems nearly impossible. Taking the ENCODE standards, a minimum **fraction of reads in peaks of 0.2** is presented but only 35 of the 210 libraries pass this threshold. Digging into Sup Table 4, one also finds that (a) the authors have used an average of 13 **PCR cycles** to amplify their libraries which is likely 5-6 more than should be necessary and (b) **some samples had as few as 270 nuclei recovered**. Needless to say, I have significant reservations about the quality of the ATAC-seq data. This is confirmed by the few times the authors show sequencing tracks, the background levels of transposition are uncommonly high (see Fig. 1c, ForBr, IRX3 and GABRA6 panels).

1.1 Response

We acknowledge the lack of clarity in Supplementary Table 4 (i.e. ATAC-seq QC metrics) and agree with the Reviewer that the range of values for multiple metrics, including FRiP, does not look plausible. This is because we calculated those metrics on per-sample peaksets rather than consensus peakset. Therefore, due to the relatively strict threshold for per-sample peak calling (FDR < 0.01), some samples with lower sequencing depth and higher background noise reported rather small peaksets that contained only peaks with the strongest signal overall which are, mostly, promoters. This led to some extremes mentioned by the Reviewer, e.g. 82% of reads in promoters. We **updated the set of reported QC metrics in Supplementary Table 4** to contain only those calculated on consensus peaksets and, additionally, we added TSS enrichment requested by the Reviewer. Additionally, we inspected all ATAC-seq samples in IGV browser and **removed additional samples** with the lowest TSS enrichment and/or less than 1,000 nuclei.

To better assess the concerns about the data quality, we **compared our data with four publicly available ATAC-seq studies** generated on postmortem human brains (raw data from those studies were processed by our computational pipeline to guarantee comparable results):

- Fullard et al 2018 (115 FANS-sorted samples from 14 regions; PMID: 29945882)
- Hauberg et al. 2020 (47 FANS-sorted samples from 3 regions; PMID: 33149216)
- Rizzardi et al. 2017 (22 FANS-sorted samples from DLPFC and NUC; PMID: 30643296)
- Bryois et al 2018 (248 bulk tissue samples from DLPFC; PMID: 30087329)

We found our data to have comparable quality based on multiple QC metrics (**Fig. S5**).

We appreciate the Reviewer's concern about data quality. However, we note that our experiments were performed on nuclei extracted from frozen postmortem brain specimens. Such tissues are, by their very nature, suboptimal due to both the effects of freeze/thawing and the post-mortem interval. This would be expected to have a direct and adverse effect on data quality when compared to fresh tissue/cell lines. This can explain a rather high average number of PCR cycles (12.5) which is, however, comparable to the other studies, including our previous open chromatin atlases (mean=14.0 [PMID: 29945882], 13.4 [PMID: 33149216]), as well as an external study (mean=11.6 [PMID: 34429139]; more comprehensive comparison with external datasets of the same experimental design is problematic since most studies do

not publish details about PCR amplification). Since PCR amplification influences the fragment length distribution [PMID: 2461560] and introduces other biases [PMID: 32213349], we compared the fragment length distributions of samples from our cohort. Despite some visual differences, they all followed a stereotypical distribution with a majority of short fragments (under 100 bp) followed by a tail of longer fragments (over 147 bp) in multiples of the nucleosomal unit size (**Fig. S6**). To further prove that we can alleviate the impact of varying levels of background noise within our data, we generated *voom plots* [PMID: 24485249] before and after adjustment for covariates (Figure R_voom; covariate selection process is described in Supplementary Methods). We were able to decrease technical variability to a level comparable with the other datasets with the same experimental design. Importantly, covariate adjustment did not remove the variance attributed to brain region differences. On the contrary, the variance attributed to the brain region differences actually increased, from 26% to 32% in neuronal samples, and from 5.6% to 8.2% in glial samples (**Fig S7**). As expected, we identified GC content-derived variables to be the main confounders in our data. GC content represents a common proxy to PCR amplification bias and Tn5:nuclei ratio ([PMID: 32213349, <https://doi.org/10.1016/j.cels.2020.02.009>]; correlation between TSS enrichment and GC content: *spearman correlation coefficient*=0.77, P-value < 2.2E-16).

Revised supplementary information:

Fig S5 | Comparison of ATAC-seq Quality Control Metrics with published results. Boxplots showing the quality control metrics between our results (current study), and published data sets (Hauberg et al., 2020¹⁰, Fullard et al. 2018⁶, Rizzardi et al., 2017⁹) generated on postmortem human brains by a similar assay, i.e. FANS ATAC-seq or homogenate ATAC-seq. **a**, TSS enrichment for housekeeping genes. **b**, PCR bottleneck coefficient reflecting the complexity of the library (ENCODE quality metrics, <https://www.encodeproject.org/data-standards/atac-seq/atac-encode4/>). **c**, Fraction of chrM reads; note that the higher values for Bryois et al. are caused by their use of whole tissue and not sorted nuclei. **d**, Fraction of uniquely mapped reads by STAR aligner. **e**, Number of “final” paired-end reads, i.e. after filtering non-uniquely mapped reads, duplicated reads and chrM reads. **f**, Fraction of reads in consensus peaksets. Note that each study has a different consensus peakset. **g**, Mean GC content. **h**, Fraction of duplicated reads. **i**, Ratio of short fragments (50-100bp) to long fragments (150-200bp) used for detecting under- or over-transposition.

Fig S6 | Fragment length distribution of the ATAC-seq data. Averaged fragment length distribution. Samples were assigned to ten bins, from the lowest to the largest short-to-nucleosome fragment count ratio (calculated as the ratio of short fragments (50-100bp) to long fragments (150-200bp)).

Fig S7 | Covariate selection and statistical modeling for ATAC-seq analysis. a-d, Violin plots of the percentage of variance explained by each covariate plus residuals over all neuronal (a, c) and non-neuronal (b, d) samples, before (a-b) and after (c-d) covariate adjustment. e-f, Tabular overview of variance before and after adjustment for covariates for neuronal (e) and non-neuronal (f) samples. g-h, Peak-wise means and variances of neuronal (g) and non-neuronal (h) samples after covariate adjustment. Peaks are represented by black points with LOWESS trends shown in red.

2. The authors claim to have data from 25 brain regions across 6 individuals. Although they are able to claim that on average 18/25 regions per individual are represented in their ATAC-seq data, this obscures the fact that one individual (BB3) only has 7 total ATAC libraries that passed filter (Supp Table 3 and 4). It is odd that there is a discrepancy between the number of RNA libraries and ATAC libraries obtained for that individual, and given the lax cutoffs for QC in the ATAC libraries, this calls into question the quality of the RNA libraries as well (as RNA-seq data has less-clear QC metrics in general). There are also differences in the number of libraries per brain region. (Supp Table 3 and 4). This is especially notable for the Globus Pallidus (GP) region which only has 1 neuronal library, and yet is 1 of only 3 brain regions analyzed as “Basal Ganglia”.

1.2 Response

We understand the Reviewer's concern about the differences in the number of ATAC-seq and RNA-seq samples. Frozen human brain is inherently a suboptimal starting material for molecular assays. In addition, the tissues used in this study were provided as thin sections and, as such, may have been more sensitive to freeze-thawing in transit which may have impacted data quality. We initially considered keeping only those samples that passed quality control for both assays. Nevertheless, we inspected RNA-seq samples that would be removed because of insufficient quality of their matching ATAC-seq counterparts and found those samples were not significantly different compared to the rest of the dataset (Fig R_qc_rna, see the first three columns in each subpanel). As such, we chose to include them in our analysis, explaining the discrepancy between the number of RNA-seq libraries and those derived from ATAC-seq. It appears that RNA-seq may be a more robust approach than ATAC-seq. To better assess the concerns about the general data quality, we compared our data with three publicly available RNA-seq studies generated on postmortem human brains (raw data from those studies were processed by our computational pipeline to guarantee the comparable results):

- Dong et al 2022 (93 FANS-sorted samples from 5 regions; PMID: 36163279)
- Hoffman et al 2019 (1,818 bulk tissue samples from DLPFC and ACC; PMID: 31551426)
- Rizzardi et al. 2017 (20 FANS-sorted samples from DLPFC and NUC; PMID: 30643296)

Despite markedly lower RIN values for the tissue used in this study, we found that our data show comparable quality based on multiple QC metrics (**Fig. S3**).

Fig S3 | Comparison of the RNA-seq Quality Control Metrics with published results. Boxplots showing the quality control metrics between our results (all samples, ATAC- & RNA-seq samples available, only RNA-seq available samples), and published data sets (Dong et al., 2022⁷, Hoffman et al. 2019⁸, Rizzardi et al., 2017⁹) generated using postmortem human brains by similar assay, i.e. FANS RNA-seq or homogenate RNA-seq. **a**, Fraction of uniquely mapped reads by STAR aligner. **b**, Number of uniquely mapped pair-end reads. **c**, Mean GC content. **d**, Ratio of short fragments (50-100bp) to long fragments (150-200bp) used for detecting under- or over-transposition. **e**, Median insert size between paired reads. **f**, Intronic rate. **g**, Intergenic rate. **h**, RNA integrity number.

3. Some of the analyses presented in main figures seem tangential to the actual central focus of the manuscript and occasionally represent results that have been known previously and may be better suited as supplemental panels. For example, the majority of Figure 3, especially figures 3c and 3d, present unsurprising findings that enhancers explain more of the variance in gene expression than do promoters.

1.3 Response

We thank the reviewer for the constructive suggestion. We agree that the second part of Figure 3 is less central to the main focus of this manuscript, and not very informative. We removed that section of analysis in the manuscript.

4. The paper presents a slew of enrichment analyses which at times are challenging to follow and the take-home messages from many of these enrichment analyses are either unclear or confirm known/published findings. Many of these analyses themselves are highly unclear. As enrichment is entirely dependent on the set of features and the background, these details matter greatly. For example, the authors write that “promoter-isoform-specific DEGs have a markedly higher enrichment for common risk variants for multiple neuropsychiatric traits” but do not explain what “enriched for common risk variants” actually means. Where does the risk variant fall with respect to the OCR? To the DEG? Each enrichment analysis should explicitly reference what is being used as their list of features and the background set of features. In analyses of enrichment, are the authors using all genes as the background set?

1.4 Response

We thank the reviewer for the comments. Throughout the manuscript, we utilized two independent methods, Stratified LD-score regression (for open chromatin) and MAGMA (for genes), to perform the genetic enrichment analysis. LD-score regression utilized the baseline model that included 24 publicly available main annotations as the background; MAGMA utilized the genome-wide protein-coding gene, both as background and controlled for the LD structures. As such, in these two analyses, we used the default background. For other enrichment analyses, such as pathway and cell type enrichment, we used all expressed genes relevant to the specific group as background. For example, for genes specific to BasGan neurons, we used all genes expressed in neuronal samples as the background set.

In the alternative promoter-isoform analysis, due to the low coverage, it's not possible to run LD-score regression on the alternative promoter OCRs. We instead assessed the genetic enrichment of the genes regulated by such promoters. While this analysis cannot distinguish if the promoter OCR or other elements contribute to the disease liability, it can be used to suggest that genes with alternative promoter-isoforms do have a higher probability of being associated with disease liability.

Given the constructive comments from the reviewers and careful consideration, we have shifted our focus to promoter-isoform analysis across brain regions, leading us to omit the co-expression/accessibility analysis section. We have performed brain region-specific enhancer-promoter links at promoter-isoform resolution. We found the distal OCRs that are specifically linked to non-5' promoters are also enriched for risk variants for neuropsychiatric disorders. And expanding the enhancer-promoter links to promoter-isoform resolution greatly increases our power to fine-map the risk variants. (new Fig. 4-6)

Revised main

More importantly, compared to brain region-specific DEGs, the genes harboring promoter-isoform-specific DEGs have a markedly higher enrichment for common risk variants for multiple neuropsychiatric traits, including SCZ (**Fig. 3d**).

5. The module-based analyses seem dubious to me. The modules that the authors focus on most are gene_1 (2454 genes), gene_18 (2732 genes), pr_7 (7271 promoters), and en_4 (3274 enhancers). Is it entirely coincidence that gene_1 and gene_18 are more than 2 standard deviations away from the mean module size and represent the two largest gene modules? This is even more striking for pr_7 which is 3 times larger than the average promoter module.

1.5 Response

We appreciate the insightful observations raised by the reviewer. Taking the reviewer's concerns into account, we acknowledge that our initial approach focusing on the co-expression network may not have been as informative as intended. Consequently, we have revised our manuscript to exclude this section. Our revised focus centers on identifying brain region-specific and promoter-isoform resolution in enhancer-promoter links. By leveraging these links, we have refined our methodology to fine-map risk genes associated with neuropsychiatric traits, as detailed in Section 1.4 and throughout the updated manuscript.

In terms of network analysis, we had also performed enrichment analysis with the top 500 genes (based on KME) for gene modules, and found the selected gene modules were still the most significant ones (not shown here). One potential reason is that the genes expressed in specific brain regions are prone to aggregation into modules that may appear larger than average. For chromatin co-accessibility analysis, a small module will hinder the achievement of statistical significance in LDSC tests.

6. I do not follow the logic of results presented on pr_7 in Fig. 5 and I'm not convinced that the overall conclusion is true. As the pr_7 module forms the basis of the last 3 figures of the paper, this warrants significant attention. The authors eventually argue that "In contrast to the other three modules, the top enriched pathways for pr_7 (Fig. 5a, and Fig. S8c) were related to regulatory functions, including the RNA processing, mitochondria, and catalytic function, none of which are significantly associated with SCZ common variants". But as far as I can tell, Fig 5b shows pr_7 having a significant enrichment in post-synapse modules which are part of "the top 50 biological pathways that are enriched for SCZ common variants". **Are the authors making a biological distinction between pathways and modules?** The claims of pr_7 being independent of cell type are equally confusing as this module comes directly from promoter elements which are well known to not be cell type-specific. Additionally, **not all disease associations must affect a single cell type and the genes included in pr_7 (at least the top 25) seem more broadly expressed.** So arguing that this is "independent of known cell types" seems contrived. Moreover, pr_7 is composed of 7,271 promoters representing nearly 1/3 of all the genes in the genome. The idea that 1/3 of the genes in the genome form a biologically coherent module is misguided. It seems entirely possible that this module only shows enrichment due to **some artifact such as it contains most of the genes expressed in brain tissue** which are enriched compared to the background set of all genes. Additionally, it seems reasonable to expect that some associations for SCZ might be related to **cell types not analyzed here.**

1.6 Response

We appreciate the detailed scrutiny by the reviewer regarding our presentation and interpretation of the results related to the pr_7 module in Fig. 5. Upon reevaluation of the reviewer's concerns, we have removed the whole co-expression/accessibility section. Our revised focus centers on identifying brain region-specific and promoter-isoform resolution in enhancer-promoter links. By leveraging these links, we have refined our methodology to fine-map risk genes associated with neuropsychiatric traits, as detailed in Section 1.4 and throughout the updated manuscript. This shift in focus allows us to provide a

more nuanced understanding of the genetic architecture underlying neuropsychiatric conditions, aligning better with the current scientific consensus and addressing the limitations previously identified.

Minor Suggestions

7. Introduction - "The prevailing view is that non-coding risk variants exert their effects by altering CRE function and disrupting gene-regulatory circuits." This statement ignores/minimizes possible effects on noncoding RNAs (something that the authors talk about later), splicing QTLs, or unannotated translated regions.

1.7 Response

We thank the reviewer for this comment and have rewritten the sentence accordingly.

Revised main:

The prevailing view is that non-coding risk variants exert their effects on gene regulation through various mechanisms, including altering cis-regulatory element (CRE) function, affecting noncoding RNA activities, and influencing splicing mechanisms.

8. It seems odd to group the limbic structures (hippocampus and amygdala) with the other forebrain structures given their clear dissimilarity.

1.8 Response

We are grateful for the reviewer's observation regarding the classification of limbic structures alongside other forebrain regions. Our original manuscript to group the hippocampus and amygdala with the neocortex was informed by two key considerations: i) Previous research, specifically the study by Kang et al. (2011) in Nature (PMID: 22031440), highlighted associations between limbic areas and the neocortex, suggesting a functional linkage that merited their grouping as a broad brain region in our analysis. ii) Our pi1 clustering analysis (illustrated in **Fig. 1e** and the newly added **Fig. S8e**) further supported this association, showing a close relationship between the limbic areas and the neocortex within our data set.

We acknowledge the reviewer's point regarding the distinct nature of the limbic and neocortical regions. To address this, we delineated both broad brain regions (including the forebrain, midbrain, and diencephalon, along with the basal ganglia) and more specific subregions (neocortex and limbic, among others). In addition, in subsequent analyses that delve into alternative promoter-isoforms (shown in the new **Fig. 3**) and enhancer-promoter links and fine-mapping (depicted in new **Figs. 4-6**), we have explicitly differentiated the neocortex and limbic regions instead of aggregating them under the forebrain category.

9. The authors state "given the challenge of separating neuronal from non-neuronal nuclei in HindBr, we provided HindBr RNA-seq and ATAC-seq as a resource but did not include these data in downstream analyses". While I appreciate the honest nature of this claim, I feel this warrants modification of the claims in the paper to have profiled 25 regions. If the authors are not confident in this data, I think they should modify the number to 24.

1.9 Response

We appreciate the reviewer's comment regarding the inclusion of the hindbrain (HindBr) in our study and acknowledge their concern about claiming the profiling of 25 brain regions. However, we believe the HindBr data still represents a valuable resource for the community, particularly since sorted HindBr

resources are scarce. Our decision to include HindBr in the total count is based on its utility as a dataset, despite the limitations in its use for certain analyses.

10. Methods on ATAC-seq peak calling and peak merging are not presented, making it challenging to evaluate the suitability of the OCR regions used. While the enrichment analyses presented in Fig S3c provide a reasonable validation, I would encourage the authors to include these details in the methods.

1.10 Response

We thank the reviewer for the comment. We had listed ATAC-seq peak calling and merging in the supplementary methods, we have now moved it to the method.

Revised methods.

Reads from the same brain region and cell type were subsequently subsampled and merged, creating 50 BAM files with a uniform depth of 170 million paired-end reads. For neuronal samples from the GP, HAB, VTA, and DRN, we had less than 170 million pair-end reads (39, 123, 136, and 162 million, respectively), so retained all reads from the corresponding samples. With the exception of those samples, all subsampling ratios were calculated per each sample individually within the 50 respective groups (brain region by cell type) to ensure that each contributes the same number of reads regardless of their overall read counts. bigWig files were created using these BAM files and peaks were called with MACS2⁹⁶. After removing peaks overlapping ENCODE blacklisted regions of anomalous, unstructured, or high signal in functional genomics assays⁹⁷, 320,308 and 196,467 peaks remained for neuronal and non-neuronal datasets, respectively. For each peak, we assigned the closest gene and the genomic context of an ATAC-seq OCR using ChIPSeeker⁹⁸; the transcript database was built by GenomicFeatures⁹⁹ upon ENSEMBL genes. Finally, read counts of all samples were quantified within these peaks using the featureCounts function in RSubread¹⁰⁰.

11. In Fig S3d, the authors present MDS plots of the RNA and ATAC data. Could the authors comment on their use of MDS rather than more conventional PCA? While I have nothing against MDS and it is a perfectly suitable method for dimensionality reduction, the methods do not detail how / what type of MDS was performed. I'm always skeptical to see such analyses without seeing the methods because I have trouble assessing whether this was performed because PCA didn't yield the desired results.

1.11 Response

We are grateful for the reviewer's insightful observation regarding our use of multidimensional scaling (MDS) in Fig S3d. We apologize for the oversight in not detailing the MDS methodology within our methods section. To clarify, we employed classical MDS using the cmdscale function in R (version 4.0.3), utilizing Euclidean distances with the default parameters set by the function.

We acknowledge the reviewer's point that PCA is a more conventional method for dimensionality reduction. In response to this feedback, we revisited our analysis and opted to replace the MDS plots with PCA in the revised figure (new **Fig S8d**). This change did not alter the results, and we obtained similar patterns of data clustering with PCA as we initially observed with MDS.

Fig. S8d, Clustering of the individual samples for ATAC-seq (N=202) and RNA-seq (N=265) using Principal component analysis. The value within the parenthesis indicates the percentage of variance explained.

12. “Consistent with previous transcriptional regulation models, we observe a high correlation between gene expression and promoter chromatin accessibility (Fig. S3e).” I think this would better be described as moderate correlation (Pearson’s $r = \sim 0.55$) which is indeed consistent with previous reports.

1.12 Response

We thank the reviewer for this comment. Considering our manuscript’s focused reevaluation on finer brain regions, we have decided to remove this specific section of analysis.

13. “We found that, in neurons, 87.5% (19,151/21,878) of genes and 79.6% (235,865/296,337) of OCRs are significantly differentially expressed/accessible in at least one of the pairwise comparisons between ForeBr, BasGan, and MidBr.” Does this include additional multiple hypothesis correction for the multiple pairwise comparisons being performed?

1.13 Response

We are grateful for the reviewer’s insightful query concerning the statistical methodology applied in our analysis. Initially, our approach to multiple hypothesis correction was specific to each of the three broad brain region comparisons (ForeBr, BasGan, and MidBr), without an additional layer of correction for the multiple pairwise comparisons undertaken.

In light of the reviewer’s comment, we have revised our statistical analysis to apply a global False Discovery Rate (FDR) correction across all pairwise comparisons. Upon applying this global FDR correction, we observed only minor changes in our results, suggesting that our original findings are robust to the stricter correction criteria.

Revised main.

We found that, in neurons, ~~87.5% (19,151/21,878)~~ 87.6% (19,157/21,878) of genes and ~~79.6% (235,865/296,337)~~ 79.5% (235,722/296,337) of OCRs are significantly differentially expressed/accessible in at least one of the pairwise comparisons between ForeBr, BasGan, and ~~MidBr~~MidDien (global FDR < 0.05).

14. I was unable to find the full results of the differential accessibility/expression results presented in Fig S4 in the Supplementary Tables.

1.14 Response

Given the file size limit, we have previously uploaded the DE result to the website (<https://www.synapse.org/#!/Synapse:syn35856920/wiki/618985>). We have now included the significant differential accessibility/expression to supplementary tables now (new **supplementary table 6-7**)

15. Not all Supplementary Tables are mentioned in the main text. Most appear to only be mentioned in the methods.

1.15 Response

We have now included all the supplementary tables in the main text.

16. Why are ForBr and BasGan combined in Fig 1f?

1.16 Response

In the initial analysis, given the pi1 plot, the non-neuronal ForBr and BasGan are very similar, so we do a ForBr-BasGan vs MidBr in non-neuronal analysis.

We have realized it might be misleading, and have separated them.

Fig. 1f, Cell type enrichment in brain region-specific gene (for DEG)(Skene et al. 2018) and OCR (for DAC)(Corces et al. 2020) sets. Neu: Neuron. pyramidal SS: somatosensory pyramidal cells. OPC: Oligodendrocyte progenitor cells. Odds ratio (OR). "-": Nominally significant ($P < 0.05$); "+": significant after FDR (Benjamini & Hochberg) correction (FDR < 0.05).

17. "Interestingly, neuronal BasGan-specific genes contain a significantly higher fraction of noncoding RNAs". No data tables or figures are cited for this claim.

1.17 Response

We have now included a Fisher's exact test analysis comparing the fraction of noncoding RNAs in neuronal Basal Ganglia-specific genes versus all neuronally expressed genes and other differentially expressed genes (new Fig. S9b).

Fig. S9b. Fisher exact test to examine ncRNA fraction differences between neuronal BasGan DEGs and other DEGs. The heights of the bars represent the enrichment (log odds ratio, OR), while the color indicates the significance (FDR, Benjamini & Hochberg correction, with all FDR values $< 10^{-16}$).

18. “To validate our annotation of the non-5’ major promoters, we utilized independent promoter-associated epigenomic data (including ATAC-Seq, and H3K4me3 and H3K27ac ChIP-Seq), around both the 5’ promoter and the non-5’ major promoter, of relevant genes in neurons”. It is unclear what is meant by “relevant genes” here. Minimally, the authors should include a list of what they’re considering “relevant genes” somewhere.

1.18 Response

We apologize for the lack of clarity due to our choice of words. By "relevant genes" we were broadly alluding to those genes for which the major promoter is non-5'. Observations that were confirmed using available external epigenomic data. We have changed the text to read as follows:

Revised main

To validate our annotation of the non-5’ major promoters, we utilized independent promoter-associated epigenomic data (including ATAC-Seq, and H3K4me3 and H3K27ac ChIP-Seq) and, as expected, the major promoters (non-5’) exhibit more potent active epigenomic modifications than the 5’ promoters.

19. “In contrast to the majority of genes (Fig. S3e), the parent gene expression of these promoter-isoforms exhibits a very limited association with promoter OCRs.” How was this analysis done for the parent gene, since it has multiple promoters? Did the authors just pick the annotated/default promoter, or take some kind of average across the promoters, or something else?

1.19 Response

We appreciate the reviewer's inquiry and apologize for any lack of clarity in our description. In our analysis, we designated the 5’ promoter as the default promoter for each gene.

Revised main

In contrast to the majority of genes (Fig. S8f and Fig. S11e), the parent gene expression of these promoter-isoforms exhibits a very limited association with OCRs at the 5' promoter (Fig. 3b; top panel)

20. "promoter-isoform expression correlates well with promoter OCRs across brain regions (Fig. 2b; bottom panel)" I would argue that a Spearman correlation coefficient of 0.2-0.4 is not very well correlated.

1.20 Response

We concur that describing the correlation as "very well" correlated might overstate the strength of the association and have revised the text accordingly.

Revised main

while promoter-isoform expressions exhibit a substantially higher correlation with promoter OCRs across brain regions.

21. In Fig. 2d and S5d, the authors describe the enrichment of promoter-isoform-specific DEGs for risk variants and biological functions, putting forth an argument that this may be largely due to enrichment for synaptic functions. Is the implication from this that the subset of genes in the promoter-isoform-specific DEGs are just essential for cellular function or do the authors mean to imply that variants affecting the noncoding gene regulatory elements are in turn affecting promoter-isoform selection. The latter seems unlikely in my opinion.

1.21 Response

We are grateful for the reviewer's insightful comments. To clarify, our analysis indeed demonstrates that the promoter-isoform-specific DEGs are significantly associated with synaptic functions and with neuropsychiatric disorders. This finding underscores the potential biological importance of these genes in synaptic operations and their relevance to disease mechanisms.

Regarding the effect of noncoding risk variants on promoter-isoform selection, we acknowledge the complexity of this issue. While our data suggest an increased regulatory complexity due to the existence of promoter-isoforms, we have not directly investigated the capability of noncoding risk variants to influence promoter-isoform selection within this study. The interplay between noncoding variants and promoter-isoform selection remains a compelling topic for future research but falls outside the immediate scope of our current manuscript.

22. "We compared the Pearson correlation between gene expression and chromatin accessibility of brain region-specific EPs, shared EPs, and non-Eps". Is chromatin accessibility the sum of the enhancer and promoter, or just the enhancer (or just the promoter)?

1.22 Response

To clarify, our analysis focused on the correlation between gene expression and chromatin accessibility at individual enhancer-promoter pairs, rather than aggregating chromatin accessibility across both enhancers and promoters.

Considering our emphasis on finer brain region specificity and the constraints imposed by the limited sample size available, we have decided to remove this particular analysis from the manuscript.

23. The Pearson correlation coefficients presented in Fig 3b showing the correlation between gene expression and enhancer activity of various ABC-based EP pairs are very low and in some cases do not even show a

substantial increase in correlation with decreasing distance (ForeBr). These results are not very strong and seem to call into question whether these EP linkages can be trusted. From my understanding, the precision/recall of the ABC model should result in much better correlation.

1.23 Response

One potential explanation for these lower correlations might be the inherent differences between transcriptional and enhancer activity modalities, coupled with a lack of sufficient biological variance within broad brain regions to distinctly capture these relationships. In our recent analysis, (Kosoy et al., 2022, Nat Genet, PMID: 35931864), a similar level of correlation was observed.

Given our focused analysis on finer brain regions and the limitations imposed by the available sample sizes for detailed correlation analysis, we have decided to omit these results from our manuscript. Instead, to substantiate our enhancer-promoter linkages, we have turned to leveraging GTex eQTL data and enhancer-promoter links derived from single-cell references. This approach allows us to present a more robust and validated set of E-P connections, better supporting our study's conclusions.

Revised Methods and Supplementary Information

Validation of brain region-unique gene-enhancer links

We validate the brain region unique- enhancer-promoter links using two independent datasets, GTEx *cis*-eQTLs from matched brain regions and gene-enhancer correlation from a multi-brain region single cell atlas (BICCN)(Li et al. 2023). For GTEx *cis*-eQTLs, we intersected ABC enhancers (flanking 500bp for both up- and down-stream) with the genomic coordinates of *cis*-eQTLs and SNPs within the same LD block of the lead SNP($r^2 > 0.8$). For BICCN gene-enhancer correlated data, we intersected the ABC enhancer regions with BICCN enhancers. Then, we evaluated the validated proportion by dividing the number of validated enhancer-targets by the number of ABC links with intersected enhancers. One-tail fisher's exact test was applied to examine whether the validated proportion for one region is higher than other regions. We evaluated 5' and non 5'-links separately, which were grouped by the types of promoter-isoforms.

Fig S14 | Validation of region-unique E-P links. brain region-specific datasets from **a**, the GTEx *cis*-eQTL¹⁴, and **b**, gene-enhancer coordination from a multi-brain region single cell atlas¹⁵. The heatmap color represents the proportion of overlapped E-P links, $\text{Proportion} = (N_{\text{validation}} / N_{\text{overlap in one region}})$, see Methods. The P values were determined with the one-tailed Fisher's exact test, indicating whether the validated proportion for one region is higher than other regions.

Fig S15 | The validated proportion for 5' and non-5' links for matched regions between our data and GTEx

24. I really have struggled to understand what is being presented in Fig 3c and how to interpret those results. If I'm having trouble interpreting it, I would imagine many other readers will be similarly confused. Though I cannot make explicit suggestions as I don't understand what is being presented, the authors should endeavor to clarify this analysis.

1.24 Response

We appreciate the reviewer's feedback concerning the interpretability of Fig 3c. Our original intention with this figure was to assess the variability in gene expression explained by different genomic features, including promoters, ABC-enhancers, and other distal open chromatin regions (OCRs), and to quantify the proportion of gene expression variance attributable to each category.

Upon reflection, and in light of your feedback, we recognize that this analysis may not have been as clear or as directly relevant to the overarching narrative of our study as intended. To ensure the manuscript's clarity and coherence, we have decided to remove this section from our presentation.

25. "However, the associated promoter OCRs for gene_18 hub genes do not exhibit concordant brain-region specificity (Fig. 4b). Similarly, the hub CRE-associated genes (referred to as hub genes) from pr_7 and en_4 are not concordant with the promoters/enhancers". These data are confusing and the authors provide no interpretation for what it all means.

1.25 Response

We are grateful for the reviewer's feedback on this aspect of our study. Initially, we intended to highlight a possible divergence between chromatin accessibility at promoters and gene expression patterns. Recognizing, however, that this point may have added complexity without clear interpretive value, especially considering the limitations of our initial co-expression/accessibility analysis, we have opted to remove this section from our analysis.

26. Discussion - "In the neurons, we identified extensive gene expression and chromatin accessibility alterations across brain regions". The authors should be more careful in using the word "alteration". I don't think that cell type-specific changes in gene expression or chromatin accessibility should be considered an "alteration".

1.26 Response

We appreciate the reviewer's careful consideration of our choice of words. In light of this feedback, we have revised our manuscript to replace the term "alterations" with language that more accurately reflects cell type-specific differences in gene expression and chromatin accessibility across brain regions.

Reviewer #2:

Remarks to the Author:

In Dong et al., the author generated a comprehensive annotation of the transcriptome and chromatin accessibility profile across 25 brain subregions in both neuronal and non-neuronal cells. The authors demonstrated that both the RNA-seq and ATAC-seq data cluster separately based on the broad brain regions and subsequently identify gene and open chromatin regions (OCRs) specific to broad brain regions that are consistent with past findings. However, due to non-neurons failing to separate broad brain regions as clearly as the neuronal cells, as well as a greater association with risk loci for neuropsychiatric traits in neuronal cells, the author focused the majority of their subsequent analysis exclusively on neurons. Firstly, the author explored promoter isoform differences between brain regions and demonstrated that they have higher enrichment for neuropsychiatric traits than all differentially expressed genes. Secondly, the authors leveraged the activity-by-contact model to label promoter-enhancer links for each broad brain region and subsequently used a variance component model to show the importance of enhancers in explaining regional differences. Thirdly, the authors generated gene co-expression analysis networks and identified four modules enriched for common risk variants for neuropsychiatric disorders that also displayed brain-region-specificity. Finally, the authors focus on the pr_7 module and show that it is associated with SCZ as well as depleted of cell-type-specific genes. Moreover, the authors use machine learning to highlight variants that overlap with pr_7 predicted to have a functional consequence.

The manuscript provides a good resource and identifies transcriptional and chromatin accessibility changes across broad brain regions. Given that single-cell level RNA-seq and ATAC-seq data is already covered in several publications (PMID: 34616060, 34390642, 34774128 etc.), especially from the prefrontal cortex, and the strategy authors using (NeuN staining) can't distinguish many of the subtypes within the neuron, and non-neuronal cell types, the real advantage of this study is to compare the datasets from 25 different subregions. Unfortunately, the current analyses, or at least what are being described in the manuscript, are at a resolution of broad brain regions with a focus only on mixed neuronal cell types, which led to the results mostly duplicating previous findings.

2 Response

We appreciate the reviewer's comments and the opportunity to clarify the unique contributions of our study. We acknowledge that much of the existing single-cell RNA-seq and ATAC-seq research, including significant work on the prefrontal cortex, has provided valuable insights into the transcriptional and chromatin accessibility landscapes of the human brain. However, our work uniquely emphasizes the subcortical brain regions—specifically the midbrain, diencephalon, and basal ganglia. These areas are pivotal to understanding neuropsychiatric disorders yet have faced experimental challenges that have limited comprehensive multiomic investigations, especially when compared to cortical areas.

For instance, the recent groundbreaking work by the BICCN, as reported by Siletti et al. (2023, Science, PMID: 37824663), represents a significant step forward in mapping the single-cell transcriptome across the adult human brain, including subcortical areas. This study found the neurons from midbrain and

diencephalon form a splatter cluster, partly due to the high heterogeneity within these neuron populations (Siletti et al 2023, Science, PMID: 37824663), underscoring the complexity and necessity of further exploring these regions at a fine-grain level. Additionally, the prevalent use of 10x single-cell reagents, which predominantly capture the 3' ends of genes, limits the resolution to isoform-level expression analysis, a gap our study aims to address. Furthermore, our research brings attention to chromatin accessibility in subcortical regions, which have been notably underrepresented in the literature. We have revised our manuscript and focus more on the fine brain regions according to the reviewer's constructive comment, and found the subcortical area exhibit high variation across fine brain regions. In response to the reviewer's constructive feedback, we have meticulously revised our manuscript with an enhanced focus on the fine brain regions. This revision revealed that the subcortical areas demonstrate significant variability across the fine brain regions. By focusing on these crucial but understudied areas of the brain, our study not only complements existing datasets from cortical regions but also fills a significant gap by providing insights into the molecular mechanisms within subcortical areas.

In our detailed examination of enhancer-promoter interactions, specifically at the level of promoter-isoforms, we discovered that distal open chromatin regions (OCRs) linked to non-5' promoters demonstrate a notable heritability for neuropsychiatric disorders on a per-SNP basis. This targeted analysis enabled us to identify links between risk variants and their target genes—connections that elude conventional analysis methods. This advancement sheds light on the complex regulatory networks at play in neuropsychiatric conditions, offering fresh perspectives on their genetic and molecular foundations.

We believe our manuscript, therefore, offers a valuable resource towards understanding transcriptional and chromatin accessibility changes across a broad spectrum of brain regions, with a particular focus on the subcortical areas implicated in neuropsychiatric diseases—regions that have been less explored in the context of multiomic investigations.

1. While the data is based on 25 distinct human brain regions and the title also emphasizes the importance of 25 distinct human brain regions, most results are at the general brain region levels but not at subregion levels. Do the authors ever try to analyze (e.g., alternative promoter isoform, enhancer-promoter links, and identify susceptible genes) at the brain subregion level?

2.1 Response

We are grateful for the reviewer's insightful query and have expanded our analysis in the revised manuscript to emphasize differences at the level of fine brain regions. Our investigations revealed that subcortical areas, particularly within the midbrain and diencephalon, display a remarkable degree of diversity at the subregion level (new **Fig. 2**). Conversely, the neocortex shows comparatively less variance across its different areas, underscoring the importance of focusing on subcortical regions for detailed brain region-specific studies (please see 2.2 response for more details).

Moreover, our detailed analysis includes differential gene expression and chromatin accessibility assessments at these finer resolutions. Notably, in the hypothalamus (ARC), we observed distinct patterns of alternative promoter-isoform usage compared to other regions within the diencephalon (**new Fig. 3b**), highlighting the substantial molecular diversity within these areas.

Further, we extended our investigation to analyze enhancer-promoter links with promoter-isoform specificity across several brain regions, including the Neocortex, limbic areas, basal ganglia, and all fine brain regions within the midbrain and diencephalon (**new Fig. 4**). By leveraging these region-specific

enhancer-promoter connections, we were able to fine-map risk variants to target genes in a manner that reflects their brain region-specific associations. (please see 2.4 response for more details).

These expanded analyses affirm the critical need and value of dissecting brain region-specific molecular mechanisms, particularly within the fine brain regions of the subcortical areas, to better understand the complex landscape of brain region diversity and its implications for neuropsychiatric disorders.

2. While most of the analysis in Figure 1 highlights the differences between broad regions clusters, it would be interesting to comment on the differences within clusters. For example, the differences between AMY, HiPP, and the rest of the forebrain regions are shown in figure 1D, specifically in the neurons.

2.2 Response

We thank the reviewer for the constructive comment. We have accordingly deepened our analysis to explore brain region specificity more hierarchically in our revisions. Beyond comparisons among broad brain regions, our enhanced analysis includes: 1) Comparisons between sub-broad brain regions, such as the neocortex versus the limbic area (AMY+HIPP) and the midbrain versus the diencephalon. 2) Within each sub-broad brain region, we examined differences against other fine brain regions, revealing notable molecular heterogeneity. Our findings indicate that despite the neocortex's substantial size and distinct cognitive functions, it exhibits limited molecular variation. The entorhinal cortex and primary visual cortex showed the most significant differences. In contrast, substantial molecular diversity was observed in the subcortical areas, especially within the midbrain and diencephalon, underscoring the necessity for detailed investigation.

Revised manuscript:

Transcriptome and chromatin accessibility are highly heterogeneous across subcortical areas

Having shown broad brain region differences, we next delved into the molecular heterogeneity across more specific regions. Despite its larger volume, the NEC displayed limited molecular changes compared to other brain regions (**Fig. 1d, Fig. S8e, and Fig. S11a**). PVC exhibits the most significant molecular differences and is enriched for excitatory neurons (**Fig. 2 and Fig. S11b**), in line with previous studies highlighting the distinct nature of excitatory neurons in PVC compared to other regions (Tasic et al. 2018). Additionally, we identified gene expression changes in the EC, with DEGs showing similarities to those found in limbic areas (**Fig. 2a and Fig. S11c**), in accordance with their anatomical position. As expected, HIPP-specific DEGs were enriched for specific excitatory neurons characteristic of the hippocampus (**Fig. 2b and Fig. S11b**).

In subcortical areas, we observed a higher degree of molecular changes across the fine brain regions. Notably, NAC and GP within BasGan exhibited substantial differences, with negatively correlated DEGs indicating molecular distinctions between the Striatum (NAC) and the Pallidum (GP) within BasGan (**Fig. S11c**). Furthermore, the GP DEGs were enriched for synaptic function and GABA B Receptor Activation, in keeping with the inhibitory effects of this region (Hallworth & Bevan 2005) (**Fig. 2b**). Within MidBr, consistent with its function, the VTA DEGs showed enrichment for dopamine neuron markers. Notably, the DACs specific to the VTA showed an enrichment of genetic risk variants associated with major depression (Fig. S11d). This aligns with prior analyses highlighting the role of dopamine neurons in the pathology of depression (Howard et al. 2019). In Diencephalon, our analysis included structures such as the hypothalamus (ARC), subthalamus (MDT), and epithalamus (HAB). As expected, the ARC DEGs were enriched for previously defined hypothalamus neuron cell makers (**Fig. 2b**). The HAB region showed enrichment for Cholesterol Biosynthesis pathways, which aligns with the role of pineal gland

neurons in producing neurosteroids (Lloyd-Evans & Waller-Evans 2020) (Fig. 2b). MDT was enriched for excitatory neuron markers (Fig. 2), and was strongly associated with neuropsychiatric genetic risk variants (Fig. S11d). Furthermore, we observed consistency between the DEGs and DACs (Fig. S11e). It is important to highlight that many of the brain regions examined in this study have not been extensively studied previously, and one novel aspect of our findings are the significant differences in gene expression and chromatin states observed within the subcortical regions.

Fig. 2 | Neuronal gene expression and chromatin accessibility changes in fine brain regions. **a**, Volcano plot of RNA-seq data from fine brain regions in neuronal cells. Significant genes after FDR (Benjamini & Hochberg) correction ($\text{FDR} < 0.05$) are marked in red. The top 5 up- and down-regulated genes for each comparison are indicated. **b**, Cell type, and pathway enrichment in fine brain region-specific gene (for DEG) (Skene et al. 2018) and OCR (for DAC) (Corces et al. 2020) sets. Neu: Neuron. pyramidal SS: somatosensory pyramidal cells. OPC: Oligodendrocyte progenitor cells. Odds ratio (OR). ".": Nominally significant ($P < 0.05$); "+": significant after FDR (Benjamini & Hochberg) correction ($\text{FDR} < 0.05$).

Fig S11a, The number of DEGs and DACs for the fine brain regions in neuronal cells.

3. While the analysis focuses on the neurons since the data is also generated for non-neurons, is it possible to identify alternative promoters in this data, or is the phenomenon specific to neuronal cell types? This also applies to all other downstream analyses.

2.3 Response

We are thankful for the reviewer's insightful query. In our study, we indeed conducted differential analyses on non-neuronal cells across broad brain regions. Interestingly, we observed that DEGs in these non-neuronal cells were enriched for neuronal cell markers and pathways characteristic of the respective brain regions. This pattern suggests 1) a potential influence from ambient RNAs and 2) the fact that non-neuronal cells, in particular abundant cell types such as oligodendrocytes, have limited brain region specificity. Given these findings, and considering the significant link between neuronal cells and neuropsychiatric disorders, we decided to focus our subsequent analyses solely on neuronal cells.

Revised main

It's worth noting that, in contrast to patterns of chromatin accessibility, DEGs in non-neuronal cells showed enrichment for markers (**Fig. 1f**) and pathways (**Fig. S10b**) characteristic of neuronal cells from corresponding brain regions. This observation could be attributed to the presence of ambient RNAs (Caglayan et al. 2022) coupled with the high degree of regional homogeneity in non-neuronal cells. Consequently, our subsequent analyses specifically focused on neuronal cells.

4. The author analyzed the alternative promoter isoform usage and enhancer-promoter links. Does the author try to combine them to identify any susceptible genes specific enhancer-isoform promoter links? If so, which is more critical for susceptible gene expression?

2.4 Response

We are grateful for the reviewer's suggestion. In response to the limitations observed in our initial co-expression/accessibility analysis, which we have since removed from our study, we delved deeper into the enhancer-promoter links at promoter-isoform resolution. Our analysis revealed that distal OCRs, specifically those associated with non-5' promoters, demonstrate a notably high per SNP heritability for neuropsychiatric disorders. Furthermore, by integrating these promoter-isoform-specific enhancer-

promoter links, we succeeded in tracing the connections from risk variants to target genes in ways that were not possible with traditional 5' promoter ABC analyses.

A case in point is the *FURIN* gene, where the 5' promoter-isoform shows minimal expression and chromatin accessibility, in contrast to the non-5' promoter-isoform, which is highly expressed (**Fig. S20**). This non-5' promoter-isoform enables the establishment of specific enhancer-promoter links to finely mapped risk variants. Such findings underscore the significance of promoter-isoform resolution, not just in isoform usage but also in the regulation of gene expression. This approach has unveiled specific enhancer-isoform promoter links associated with susceptibility genes, providing critical insights into the molecular mechanisms underpinning neuropsychiatric disorders.

Revised manuscript

Brain region-specific enhancer-promoter links at promoter-isoform resolution

Considering the majority of OCRs are distal, we aimed to identify their target genes using the activity-by-contact (ABC) model (Fulco et al. 2019), which combines enhancer activity with the spatial proximity between enhancers and promoters. Recognizing the critical role of promoters in E-P interactions and cis-regulation, and that the 5' promoter is not necessarily the most active promoter, we refined our analysis to capture enhancer-promoter connections at the promoter-isoform level (Methods). Incorporating our brain-region-specific chromatin accessibility profiling and cell-type-specific Hi-C contacts (Rahman et al. 2021; Dong et al. 2022), we established E-P links at promoter-isoform resolution including both broad brain regions such as BasGan, NEC, and Limbic, and fine regions within the MidDien, reflecting their significant molecular diversity.

Across the brain regions, we identified between 24,384 and 30,721 E-P links ($ABC > 0.02$) at promoter-isoform resolution (**Fig. S13a, Supplementary Table 11**), covering 19,486 to 24,042 enhancer-gene pairs (**Fig. S13b**), with the majorities of the links being within 50 kb (**Fig. S13c**). Notably, $28.78\% \pm 0.62\%$ (mean \pm se, **Fig. S13d**) of genes possessed multiple linked promoter-isoforms, and $38.35\% \pm 0.45\%$ of the ABC-linked enhancers were predicted to regulate multiple isoforms (**Fig. S13e**). To assess the brain region specificity of E-P links, we compared the pairwise correlation of link strength (ABC score) and the overlaps of links among different brain regions (**Fig. 4a-b**). As expected, brain regions within the same broad brain regions exhibit higher similarity. While distinct broad brain regions, such as BasGan, displayed a notable 41.3% of unique links (**Fig. 4c**), consistent with observed gene expression and chromatin accessibility variations across regions (**Fig. 1d**).

To validate the brain region-specific E-P isoform links, we leveraged independent brain region-specific datasets from the GTEx *cis*-eQTL (GTEx Consortium 2020) (**Fig. S14a**), and gene-enhancer coordination from a multi-brain region single cell atlas (Li et al. 2023) (**Fig. S14b**). Our brain region-unique E-P isoform (both 5' and non-5') links were corroborated by data from corresponding brain regions or abundant cell types within those regions. For instance, our BasGan unique E-P links were well corroborated by BasGan eQTLs and cis-coordinations of BasGan-restricted MSN (**Fig. S14-15**). These results confirmed the reliability of brain region specificity of both 5' and non-5' promoter E-P links.

Considering that the majority of neuropsychiatric disorder risk variants are noncoding, we reasoned that these variants would be associated with enhancers and promoters in the brain and exert their effects by disrupting gene regulatory circuits. As such, we assessed the heritability of disease risk attributed to different types of regulatory elements, including 5' (accounting for $18.9\% \pm 0.25\%$ of all promoter OCRs) and non-5' ($18.86\% \pm 0.34\%$) promoter-isoforms, enhancers that predicted to regulate 5' ($10.97\% \pm 0.58\%$) and non-5' specific ($5.02\% \pm 0.24\%$) promoters, and non-ABC elements (**Fig. 4d-e, Fig. S16**).

As expected, regulatory elements across brain region groups were enriched for disease risk variants, particularly for SCZ and BD (Fig. 4d-e). Notably, ABC model-predicted enhancers demonstrated significantly higher per-single nucleotide polymorphism (SNP) heritability compared to other distal OCRs (Fig. 4d). Furthermore, non-5' promoter-isoform-specific ABC enhancers showed a similar enrichment to 5' promoter-isoforms, underscoring the importance of analyzing E-P links with promoter-isoform specificity. Additionally, promoter regions involved in ABC links exhibited substantially higher disease heritability (Fig. 4e), reinforcing the significance of these regulatory connections.

Fig. 4 | The brain-region specificity of ABC enhancer-promoter links at isoform resolution. **a**, Heatmap represents the spearman correlation coefficients of isoform resolution E-P link strength (ABC score) between brain regions. **b**, Circos plot of pairwise overlaps of ABC links among brain regions. The inner color indicates the number of links. **c**, Percentage of unique E-P links for each brain region. Enrichment of common variants for different neuropsychiatric disorders in ABC. **d**, enhancers and **e**, promoters across brain regions. A positive coefficient signifies enrichment in heritability (Normalized tau). Negative coefficients were displayed with gray blocks. "+" indicates significant enrichment after FDR (Benjamini & Hochberg) correction (FDR < 0.05). The sidebars indicate brain regions and types of regulatory elements.

Fine-mapping of SCZ risk variants

Having demonstrated the strong association between genetic risk variants and the ABC enhancer of both 5' and non-5' promoter-isoforms, we further leverage these E-P links to characterize the regulatory mechanisms of SCZ risk loci, including potential causal variants, associated enhancers, and affected target genes. We focused on fine-mapped risk variants from the latest GWAS study (Trubetskoy et al. 2022) with a posterior inclusion probability (PIP) > 1%, employing the ABC-MAX strategy (Nasser et al. 2021) to pinpoint the affected promoter-isoform and genes (**Fig. 5a**). In total, we prioritized 72 genes across brain regions for 122 out of 4,319 fine-mapped SNP overlapped with ABC enhancers (**Supplementary Table 12**). In line with previous analysis (Trubetskoy et al. 2022), these targets are mainly related to synaptic vesicle budding and regulation of synaptic plasticity (**Fig. S17**). For validation, we used an orthogonal method, polygenic priority score (PoPS) (Weeks et al. 2023), enhanced by integrating recent brain-related gene features (**Methods**), which significantly improved the prioritization heritability (**Fig. S18**).

Notably, the majority ($60.82\% \pm 2.56\%$) of genes were identifiable only through non-5' specific E-P links (**Fig. 5b**) and those scoring high on PoPS (**Fig. 5c**), emphasizing the critical role of promoter-isoform specificity in E-P links. For example, the fine-mapped SNP rs7178152 (PIP = 14.53%), was linked to *ABHD2* through a non-5' promoter-isoform E-P link, despite being located nearer to the 5' promoter of *FANCI*. *ABHD2*, implicated in neurotransmitter release (Baggelaar et al. 2018; Ogasawara et al. 2016), exhibited a considerably higher PoPS score (0.87 vs *FANCI* of 0.62; rank percentage) and was nominated as a causal gene by our previous enhancer QTL-based fine-mapping (Dong et al. 2022).

Moreover, our analysis highlights the specificity of brain regions in the genetic regulation of fine-mapped genes, with 21 out of 72 SCZ prioritized genes unique to specific brain regions (**Fig. S19a**). For instance, the fine-mapped SNP rs2944829 (PIP=3.21%) overlapped a BasGan-specific OCR and was linked to *CALN1* only in BasGan (**Fig. 5e**), despite *CALN1* being expressed across various brain regions (**Fig. S20a**). The brain region-specific enhancers and linked target genes provide candidate regulatory mechanisms for enhanced disease susceptibility in certain areas.

Fig. 5 | Prioritized genes for schizophrenia fine-mapped SNPs using ABC enhancer-promoter-isoform links across brain regions. **a**, Target genes with ABC-max score for SCZ fine-mapped SNPs (PIP > 1%). The ABC target panel's yellow, blue, and light blue boxes represent target genes identified by 5' isoforms, non-5' isoforms, and both, respectively. The proximity panel indicates whether the gene is closest (blue box) to fine-mapped SNPs, or not (yellow box). Genes with the top 95% PoPS score were labeled. **b**, The proportion of genes identified by 5' and non-5' promoter-isoforms across brain regions. **c**, The distribution of PoPS score of target genes identified by 5', non-5' isoforms, and genes closest to fine-mapped SNPs. UCSC browser of ABC-MAX target isoforms of fine-mapped SNPs **d**, rs7178152, and **e**, rs2944829. The dashed line in the GWAS track represents p-value = 5×10^{-8} . The colors in the ABC E-P link track correspond with the colors of brain regions in the chromatin accessibility track. The lines in the promoter-isoform track indicate the rank of the promoter-isoforms of target genes, with the ABC-linked promoter-isoforms highlighted in red.

Fine-mapping of BD risk variants

We further applied the ABC-Max method to BD genetic variants (**Fig. 6a**), concentrating on GWAS fine-mapped SNPs (PIP > 1%). This analysis identified 31 genes linked to 46 fine-mapped SNPs (**Supplementary Table 12**), which displayed notably high PoPS scores. Mirroring findings in SCZ, a considerable fraction of BD risk variants was uniquely identified with non-5' promoter-isoforms (47.61% ± 10.65%) (**Fig. 6b-c**), in concordance with their highly active expression. For example, the SNP rs1894401 (PIP = 1.11%), linked to the *FURIN* (PoPS = 0.99) known for its role in neuropsychiatric disorders (Fromer et al. 2016; Zhang et al. 2022), is situated near the *FES* promoter (**Fig. 6d**). Notably, the 5' promoter-isoform of *FURIN* is not expressed, whereas the 2nd and 4th non-5' isoforms were highly expressed and linked to the fine-mapped SNP in VTA and NEC, respectively (**Fig. S20b**). In another case, SNP rs7622851 (PIP = 8.38%), is connected to the highly expressed non-5' isoform of the *WDR82* (PoPS = 0.98) gene across multiple MidDien regions, where the 5' isoform is minimally expressed (**Fig. S20c**). These observations suggest the importance of incorporating promoter-isoform resolution in E-P link analysis.

Extending this analysis to other neuropsychiatric traits (**Supplementary Table 12**) revealed that 61.51% ± 1.41% of targets were not closest to the lead SNP (**Fig. S21**), 43.91% ± 1.67% were exclusively linked to non-5' promoter-isoforms (**Fig. S22**), and 27.67% ± 1.04% were only involved in one brain region (**Fig. S19b**). Interestingly, the prioritized non-5' target genes showed higher PoPS scores compared to their 5' counterparts (**Fig. S23**), with no significant differences observed in ABC scores (**Fig. S24**) or distances (**Fig. S25**). This extensive mapping further underscores the necessity of considering brain region-specific and promoter-isoform-specific regulatory landscapes for a deeper understanding of genetic susceptibility in neuropsychiatric disorders, highlighting the nuanced genetic regulation within these conditions.

Fig. 6 | Prioritized genes for bipolar disorder fine-mapped SNPs using ABC E-P links at promoter-isoform resolution across brain regions. a, Target genes with ABC-max score for BD fine-mapped SNPs (PIP > 1%). The ABC target panel's yellow, blue, and light blue boxes represent target genes identified by 5' isoforms, non-5' isoforms, and both levels, respectively. The proximity panel indicates whether the gene is closest (blue box) to fine-mapped SNPs, or not (yellow box). Genes with the top 95% PoPS score were labeled. **b**, The proportion of genes identified by 5' and non-5' promoter-isoforms across brain regions. **c**, The distribution of PoPS score of target genes identified by 5', non-5' isoforms, and genes closest to fine-mapped SNPs. UCSC browser of ABC-MAX target isoforms of fine-mapped SNPs **d**, rs1894401, and **e**, rs7622851. The dashed line in the GWAS track represents p -value = 5×10^{-8} . The colors in the ABC E-P link track correspond with the colors of brain regions in the chromatin accessibility track. The lines in the promoter-isoform track indicate the rank of the promoter-isoforms of target genes, with the ABC-linked promoter-isoforms highlighted in red.

5. Line 212, the authors wrote: The recently developed activity-by-contact (ABC) model integrating chromatin states and 3D contacts accurately predicts the experimentally validated promoter-enhancer interactions. The ABC model perhaps is the only existing model for predicting E-P pairs, but it is far from being "accurate." The major drawback in the current ABC model is it used average Hi-C signal from not necessarily relevant cell lines and led to the bias towards H3K27ac in gene regulation. A recent publication (PMID: 35951677) suggested that three-dimensional chromosomal interactions play a significant role in enhancer gene regulation. Thus, cell type-specific looping information is critical for the dissection of gene regulatory mechanisms. Can the authors use cell type-specific loops published in PMID: 33057195 and 31727856?

2.5 Response

We thank the reviewer for their critical insights regarding the limitations of the ABC model and the potential for improved accuracy through cell type-specific chromosomal interactions. Acknowledging the points raised, we have revised our manuscript to eliminate the previous assertion about the ABC model's accuracy.

We concur with the notion that PLAC-seq, by providing cell type-specific insights, represents a more precise approach for capturing gene regulatory mechanisms. Nevertheless, the diversity of brain regions and cell types, coupled with the specific challenges associated with subcortical areas, presents a significant obstacle to obtaining comprehensive cell type-specific Hi-C or PLAC-seq data for these regions. Given these considerations, and the specificity of the data available from studies like those reported in PMID: 33057195 (focused on the developing brain and frontal cortex) and PMID: 31727856 (centered on the frontal cortex), we are cautious about their applicability to our study's broader scope. These datasets, while invaluable, may not be suitable for our multibrain region analysis.

6. In the following text, "As such, we constructed weighted gene co-expression analysis (WGCNA) networks with genes, promoters, and enhancers (EP-linked enhancers), to systematically examine the co-expression and co-accessibility modules implicated in disease etiology. Together, we identified 19 gene modules, 7 promoter modules, and 11 enhancer module" The author should clarify that the gene modules examine the expression, and the promoter and enhancers modules examine co-accessibility. It isn't easy to understand what data is used to identify these modules. Do the authors try to combine some of them (e.g., promoter and enhancer) to do WGCNA analysis?

2.6 Response

We appreciate the reviewer's request for clarification regarding our methodology. In our initial analysis, we separated promoters and enhancers due to Computational constraints—integrating both promoter and enhancer data into a single WGCNA analysis would exceed the capabilities of the current computational framework due to matrix size limitations. Moreover, our preliminary subsampling indicated that promoters and enhancers form distinct clusters (not shown), likely influenced by differences in GC content and the observation that enhancers exhibit more brain region-specific patterns.

Upon further reflection and considering the feedback, we acknowledge that this portion of our analysis did not yield insights as informative as anticipated. As such, we have opted to remove this section from the revised manuscript to maintain focus on the clearest and most impactful findings of our analysis.

7. The author compared results with two multi-brain region single cells data. Does the author also try to compare their subregion result with data from the Allen brain multiple cortical areas - smart-seq data?

2.7 Response

Allen Brain Institute's Smart-seq early data are generated at bulk resolution (Hawrylycz et al, 2012, Nature. PMID 22996553), where cell type heterogeneity significantly influences the results. Consequently, we initially refrained from direct comparisons due to the predominance of cell-type-specific signals in their dataset.

Nevertheless, in response to the reviewer's comment, we performed a comparison with recent single-cell Smart-seq data that encompasses both cortical and hippocampal regions (Yao et al, 2021, Cell. PMID: 34004146). We compared our differentially expressed genes from the broad brain regions (**Fig. S10a**) as well as the fine brain regions (**Fig. S11b**) and, in both analyses, found a high concordance between the brain region and expected cell type, underscoring the reliability and relevance of our results in reflecting the molecular landscape across different brain subregions.

Revised supplementary information

Fig S10a, Cell type enrichment of the ForeBr neuronal DEGs with reference from Allen Brain Institute(Yao et al. 2021). Glu, glutamatergic neuron. GABA, gabaergic neuron. IT, intratelencephalic neuron. CT, corticothalamic. DG, dentate gyrus. FC, fasciola cinereal. IG, induseum griseum. ProS, prosubiculum.

Fig S11b, ForeBr neuronal fine brain region DEGs enrichment with reference from Allen Brain Institute(Yao et al. 2021). Glu, glutamatergic neuron. GABA, gabaergic neuron. IT, intratelencephalic neuron. DG, dentate gyrus.

8. Fig.4C, only one scale of P value should be used. The author could use continuous color to indicate FDR.

2.8 Response

After careful consideration of the comment and a review of the analytical impact, we have decided to remove this figure from our manuscript.

9. The authors claimed, "we next annotated the functional effects of common variants associated with the regulatory elements identified here," which overstates the impact. They should state that they predicted the functional effects instead.

2.9 Response

We have removed the entire section including co-accessibility analysis and machine learning fine-mapping, and focus on brain region-specific enhancer-promoter links at promoter-isoforms resolution.

10. Machine learning results predict functional SNP at the risk gene (e.g., rs133376 in the NAGA gene). Does SP family TF have a high expressing level based on the RNA-seq data? It would be better if the author could further use the experiment to validate the SNP's function, leading to reliable novel findings.

2.10 Response

We appreciate the reviewer's insightful suggestions. It is indeed true that SP family transcription factors, including SP1, SP4, and KLF9, were expressed in the human brain, with SP4 being linked to rare variants associated with schizophrenia. However, we have decided to remove the section pertaining to machine learning predictions of functional SNPs. Instead, our revised analysis concentrates on brain region-specific enhancer-promoter links at the level of promoter-isoforms.

Reviewer #3:

Remarks to the Author:

Dong and colleagues set out to close what they see as a gap in the available data, pertinent to neuropsychiatric disease. They recognize that "brain region- and cell-specific transcriptomic and epigenomic molecular features are associated with heritability for neuropsychiatric traits" but perceive that a "systematic view, considering cortical and subcortical regions, is lacking".

At the outset, one might query the novelty of such a study, and significance of the advance - even given the apparent robustness of the work. Studies published in 2020 by the Shendure group (PMID: 33184180) in the application of similar methods to human fetal tissue and by **Hook (PMID: 32303558)** using layer-identified cortical neurons from mouse specifically to explore psychiatric disease establish a significant precedent for this work that appears to be neglected by the authors.

The methods used, data generated and the statistical robustness do not raise any particular concern for me. My concerns lie simply in the extent to which the observations made by the authors represent a significant advance in our understanding of the biological/cell-dependent/mechanistic/genetic basis of neurological disease. At this stage one would expect that the experiment is significant and feasible; that the observations are predictable; but that they are not supported by independent, systematic functional inquiry/validation.

In brief, I believe the abstract captures to take home message. The authors generate a significant resource but, in my opinion, fall short of taking critical additional steps to providing substantial new insight.

3 Response

We thank the reviewer for their comments and welcome the opportunity to elucidate the unique contributions and advances our study presents in the context of neuropsychiatric disease research.

Our study acknowledges the rich body of work surrounding cortical regions, particularly the prefrontal cortex, in neuropsychiatric research. However, we identified a significant gap in the comprehensive exploration of subcortical regions, including the midbrain and diencephalon, which play pivotal roles in neuropsychiatric conditions yet remain underrepresented in the literature. Our research specifically aimed to bridge this gap by providing a systematic analysis that includes both cortical and subcortical regions, thereby offering a more complete view of the brain's molecular landscape relevant to neuropsychiatric traits. Indeed, while previous studies, such as those conducted by the Shendure group (PMID: 33184180) and Hook (PMID: 32303558), have made significant contributions to our understanding of cell type specific molecular characteristics, our work extends these efforts by integrating a comprehensive analysis of both cortical and, crucially, subcortical areas. This approach allowed us to uncover distinct molecular signatures across regions of the human brain, highlighting the unique contribution of subcortical areas to function and disease.

One outcome from our study is the identification and analysis of alternative promoter-isoform usage across the human brain, which, to our knowledge, is novel. Our results reveal that the canonical 5' promoter is not always the most actively expressed, challenging the prevailing assumptions about promoter activity. Importantly, genes with alternative promoter usage were found to be strongly enriched for neuropsychiatric disorders, underscoring the relevance of our findings towards understanding the genetic basis of these conditions.

By refining our analysis to examine enhancer-promoter connections at the promoter-isoform level, we revealed that distal open chromatin regions (OCRs) associated with non-5' promoters exhibit high per SNP heritability for neuropsychiatric disorders. This nuanced approach allowed us to establish connections from risk variants to target genes in a manner not achievable through traditional analyses, providing new insights into the regulatory mechanisms underlying the etiology of neuropsychiatric disorders.

In conclusion, our study generates a significant resource by systematically documenting transcriptional and epigenomic profiles across a wide array of brain regions, including underexplored subcortical areas. Through our novel findings on promoter-isoform usage and its implications for disease, we believe our work represents a meaningful advance in unraveling the complex biological and genetic underpinnings of neurological diseases. We acknowledge the reviewer's perspective on the necessity for further functional validation and view our study as an important step towards inspiring and enabling such future investigations.

REVIEWERS' COMMENTS

Reviewer #1 (Remarks to the Author):

Dong et al. present a heavily revised manuscript that now focuses on the mapping of different promoter isoforms and the associated gene regulatory interactions that they identify by combining ATAC-seq and RNA-seq data across a large compendium of bulk datasets acquired from a diversity of brain regions. I commend the authors for such a large revision. The amount of work represented in this revision is substantial and I feel that the manuscript is much improved by the new inclusions and omissions and the more concrete focus on promoter-isoforms. Many of my initial concerns have been addressed, either by inclusion of updates or omission of analyses so I focus my comments on a small number of concerns that came up in the modified manuscript.

1. The authors make a large distinction between 5' and non-5' promoter isoforms. I acknowledge that a distinction is needed but I'm not familiar with why the 5' vs non-5' distinction is used. I would find it more helpful to use a "default" vs "non-default" distinction, for example by using the promoter annotated by RefSeq vs alternative promoters. The reason I mention this is that in the current distinction, if the authors find that a non-5' promoter is used, I'm not sure how to assess that because if that non-5' promoter is still the default promoter, then the authors results are less novel than they would be if the non-5' promoter was something else / something new. If my interpretation of this is mis-informed and in fact the 5' isoform is actually the default isoform, then this concern is not valid; however, that is not my understanding.

Response.

Thank you for your comment. The 5' promoter is used as the default promoter in our analyses (just as most the studies use). To clarify, we have added to the main text that the 5' promoter is often assumed to be the default promoter. However, we found that for a significant number of genes (1,344 out of 11,224), the non-5' promoter-isoforms are the major promoters (Supplementary Fig. 12a). This highlights the importance of examining alternative promoters beyond the default.

Main text.

Although without regulatory genomics annotation, the 5' promoter is often assumed as the default promoter, we found, at least for a fraction of genes (1,344/11,224), that the 5' promoter-isoform is not the most highly active (i.e., major promoter, **Supplementary Fig. 12a**).

2. While this is not my specific expertise, I am not used to PIP being presented as a percentage but rather a fraction. And in that capacity, I am used to seeing PIP scores greater than 0.9 for likely causative fine-mapped variants. However, the examples that the authors use have PIP scores less than 15%. Does this correspond to a PIP less than 0.15 or is this metric somehow inverted and it represents a PIP greater than 0.85? Can the authors clarify this discrepancy and (if relevant) justify why their PIP scores appear to be so low? It would certainly seem outside of the standard in the field to be highlighting variants with a PIP less than 0.15. Again, not my expertise but this seems like a nonstandard presentation and it might behoove the authors to use the more standard presentation of PIP.

Response.

Thank you for your comment. We apologize for any confusion. In our analyses, PIP values are fractions between 0 and 1 (e.g., a PIP of 14.53% is 0.1453). Due to complex linkage disequilibrium (LD) in neuropsychiatric traits, causal probability is distributed across many SNPs, leading to lower individual PIP scores. High PIP values (>0.9) are uncommon in this context. We used a PIP threshold of >0.01 to identify fine-mapped SNPs, aligning with current standards.

3. For the fine-mapped examples presented in Figs. 5d-e and 6d-e, could the authors provide additional layers of information that might lend more confidence to these being the causal SNPs? For example, if these variants are thought to be acting by perturbing gene regulation, do they reside within a transcription factor motif and is the base change something that would be damaging to TF binding?

Response.

We appreciate the reviewer's thoughtful comment and interest in further exploring the potential mechanisms by which the fine-mapped SNPs may influence gene regulation. Our study focuses on the genetic fine mapping of causal genes by identifying enhancer-promoter interactions at the promoter-isoform level across various brain regions. The primary goal was to establish a comprehensive resource that highlights the importance of promoter-isoforms and regional specificity in understanding the genetic architecture of neuropsychiatric disorders. While we agree that assessing whether these fine-mapped SNPs reside within transcription factor (TF) binding motifs and evaluating the impact of allelic changes on TF binding could provide additional evidence supporting their causal roles, such analyses are beyond the scope of the current study. We have acknowledged it in our discussion section now.

Revised main text

While our promoter-isoform resolution genetic fine-mapping successfully identified genes that are missed by gene-level analyses and single brain region approaches, we did not investigate the specific mechanisms by which individual fine-mapped SNPs may influence gene regulation. Future studies are warranted to assess whether these SNPs disrupt the regulatory machinery, such as transcription factor binding motifs, and how they exert their functional effects.

4. Related to 1.2 Response, I think it would be appropriate for the authors to modify how they discuss the number of samples obtained in the study to accurately capture the difference between the total number of samples assayed (25 brain regions across 6 individuals) and the actual number of pass-filter samples. However, I know that the authors have already made their preference clear on this matter and they have also removed the number of brain regions from the title so I leave this suggestion to editorial discretion.

Response

Thank you for your insightful comment regarding the presentation of sample numbers in our study. We agree that it is important to clearly distinguish between the total number of samples assayed and the number of samples that passed quality control filters. In response to your suggestion, we have added details in the main text to clarify the total number of samples collected from the 25 brain regions across six individuals and the number of samples that passed quality control and were used in downstream analyses.

We have also removed the reference to "25 brain regions" from the title to prevent any confusion.

Revised title.

A Multi-Regional Human Brain Atlas of Chromatin Accessibility and Gene Expression Facilitates Promoter-Isoform Resolution Genetic Fine-Mapping

Main text.

After rigorous quality control, including assessing cell type, sex, genotype concordance, and quality metrics (**Supplementary Fig. 2-S7** and **Supplementary Notes**), we obtained a total of 14.2 billion uniquely mapped paired-end read pairs for RNA-seq (N=265), and 15.1 billion uniquely mapped paired-end read pairs for ATAC-seq (N=202).

The HinBr exhibits the most significant differences in comparison to the other brain regions. However, given the challenge of separating neuronal from non-neuronal nuclei in HindBr²⁴, we provided HindBr RNA-seq and ATAC-seq as a resource but did not include these data in downstream analyses.

Reviewer #3 (Remarks to the Author):

The authors base their response to questions of novelty and significance on the observed "alternative promoter usage" but provide no response to the question of functional validation. In the absence of orthogonal validation, I worry that the observation stands without biological meaning.

Response.

Thank you for your comment. To address the question of functional validation, we provided orthogonal evidence by integrating ATAC-seq data (chromatin accessibility) at promoter loci with our RNA-seq data (Figure 3). We demonstrated that promoter-isoform-specific open chromatin regions correlate strongly with the expression levels of the corresponding promoter-isoforms across brain regions. This correlation structure suggests that the observed alternative promoter usage is functionally relevant and biologically meaningful. While we acknowledge that additional functional assays could further validate these findings, such experiments are beyond the scope of the current study.